# Actin waves guide an outward movement of microclusters in the lymphocyte immunological synapse

Aheria Dey [1,4], Samuel Z Khiangte [2,4], Srishti Mandal [1], Huw Colin-York[3], Marco Fritzsche[3], Sumantra Sarkar [2✉] & Sudha Kumari [1✉]

## Abstract

**The lymphocyte immune response begins with antigen recognition on antigen-presenting cells, leading to the formation of the immunological synapse—a specialized interface for biochemical and biophysical exchange. At the synapse, most antigen-engaged receptor microclusters move inward toward the central supramolecular activation cluster (cSMAC) via retrograde F-actin flow, eventually clearing from the cell surface. This retrograde movement and receptor downregulation maintain antigen receptor homeostasis, critical for adaptive immunity, though its regulation remains unclear. Using live T cells, we identify a significant pool of antigen-engaged microclusters moving anterogradely toward the cell periphery, rather than the cSMAC. This movement is driven by actin waves propagating outward and coupling to microclusters through the Wiskott-Aldrich Syndrome Protein. These findings reveal a previously unrecognized mode of actin dynamics—anterograde actin waves—that co-exist with retrograde flow and direct microclusters away from the downregulation zone. This dual actin behavior underscores the complex cytoskeletal mechanisms T cells employ to regulate receptor distribution and maintain signaling homeostasis during immune activation.**

**Keywords** Actin Retrograde Flow; T Cell Receptor Microcluster Dynamics; Actin Waves; Wiskott-Aldrich Syndrome Protein; Immunological Synapse
**Subject Categories** Cell Adhesion, Polarity & Cytoskeleton; Immunology

## Introduction

The formation of a specialized cell-cell contact interface between the T cells and antigen-presenting cell (APC), the immunological synapse, signifies the initiation of adaptive immune response. Synapse allows the engagement of receptors and ligands between the cell surfaces, enabling molecular recognition events, including the engagement of T cell receptors with ligands on the APC, such as major histocompatibility complex molecules loaded with agonist peptides. The engaged T cell receptors serve as signaling units and form microclusters that translocate within synapses (Courtney et al, 2018; Varma et al, 2006; Dustin, 2008). In its mature form, the synapse exhibits a bull's eye organization with three distinct radially concentric zones (Dustin, 2009b; Monks et al, 1998) termed central supramolecular activation zone (cSMAC), peripheral supramolecular activation zone (pSMAC), and distal supramolecular activation zone (dSMAC), respectively. The T cell receptor engagement and activation occur in the dSMAC and pSMAC zones and the engaged receptors then migrate in a centripetal fashion traversing pSMAC to finally accumulate in cSMAC (Varma et al, 2006). At cSMAC, a major fraction of the antigen receptor exists in extracellular vesicles, with little or no signaling. The delivery of the antigen receptor to signaling poor cSMAC serves a crucial role in antigen receptor desensitization as well as extracellular communication (Saliba et al, 2019; Kim et al, 2018; Stinchcombe et al, 2023; Choudhuri et al, 2014). Overall, the sub-synaptic positioning as well as dynamics of engaged receptors in SMACs determine some of the essential qualitative features of the T cell immune response (Varma et al, 2006; Campi et al, 2005; Dustin, 2009a, 2010; Hartman et al, 2009; Mossman et al, 2005).

Given its crucial importance in T cell receptor homeostasis at the synapse, the mechanisms of T cell receptor translocation from the periphery to cSMAC have been studied previously and found to be dependent on actin organization and dynamics (Campi et al, 2005; DeMond et al, 2008; Murugesan et al, 2016). Indeed, T cell receptor microclusters' migration speed correlates with the speed of the centripetal flow of actin (DeMond et al, 2008; Murugesan et al, 2016; Kaizuka et al, 2007; Yu et al, 2010; Kumari et al, 2015), and breaking the centripetal actin flow using micropatterned substrates leads to a disruption in centripetal microcluster trajectories and the accumulation of T cell receptor at the actin flow breakpoints (DeMond et al, 2008; Yu et al, 2010).

Do all engaged T cell receptor microclusters undergo centripetal translocation and accumulate into cSMAC? While synapse imaging studies thus far indicate so, the surface expression analysis of antigen receptors using flow cytometry has shown an average of 40–60% of antigen-induced downregulation (Valitutti et al, 1997;

[1]Department of Microbiology and Cell Biology, Indian Institute of Science, Bengaluru 560012, India. [2]Department of Physics, Indian Institute of Science, Bengaluru 560012, India. [3]University of Oxford, Oxfordshire, UK. [4]These authors contributed equally: Aheria Dey, Samuel Z Khiangte. ✉E-mail: sumantra@iisc.ac.in; sudhakm@iisc.ac.in

José et al, 2000) implying that, on average, ~50% of antigen receptors escape the degradative fate. We investigated this inconsistency to examine if indeed the antigen receptor undergoes a 'directional sorting' at the immunological synapse during T cell activation. Using live Primary T cells activated on APC-mimetic reconstituted bilayers, in combination with fast super-resolution imaging, an unbiased tracking algorithm, and pharmacological inhibitors, we find that a sizeable fraction of motile clusters translocate not towards cSMAC but away from cSMAC. This motion is enabled by a previously uncharacterized wave-like anterograde flow of F-actin, which couples with a fraction of microclusters using Wiskott-Aldrich Syndrome Protein (WASP)/ WASP-interacting proteins, sweeping them away from cSMAC towards the cell periphery. These results reveal a novel cytoskeletal behavior of T cells implicating the diversity of cell-intrinsic mechanisms that mediate TCR homeostasis in the wake of an immune response.

## Results and discussion

### Primary T-cell synapses display anterogradely motile TCR microclusters

To generate planar T cell synapses that can be imaged at high spatiotemporal resolution for tracking ligated T cell receptor microcluster (referred to as "TCR" henceforth) movement, we utilized the supported lipid bilayer-based T cell activation system that has been previously used to examine subsynaptic receptor dynamics and movement (DeMond et al, 2008; Murugesan et al, 2016; Grakoui et al, 1999; Crites et al, 2015; Torres et al, 2013; Glazier and Salaita, 2017; Babich et al, 2012). Mouse Primary CD8+ T cells (see 'Methods') or the CD8+ 1G4 Jurkat T cell line (Lee et al, 2017; Colin-York et al, 2019a) were incubated with supported lipid bilayers (SLB) reconstituted with a 6X-Histidine tag containing the extracellular domain of ICAM1 and biotinylated and Alexa 568 labeled anti-CD3 agonist antibodies (Okt3 and 2C11 clones, respectively), and the SLB–T cell conjugates were imaged live using Total Internal Reflection-Structured Illumination (TIRF-SIM) microscopy (Fig. 1A; 'Methods' section). The acquired images were processed (see 'Methods') and analyzed using dynamic tracking and particle image velocimetry algorithms (Python). The image processing scheme sensitively detected microclusters within the synapses with an accuracy of 97.2% (Figs. 1B and EV1A). Detected microclusters were tracked using TrackPy to assess the magnitude and direction of all detected motile microcluster entities, and tracking accuracy was verified using independent manual tracking, where the speeds were found to be comparable (48.26 ± 8.91 nm/s using Python and 45.72 ± 6.82 nm/s using manual tracking) (Fig. EV1B, Movie EV1). Analysis of TCR tracks showed a clear net centripetal movement ("retrograde or inward fraction") in Jurkat T cells (Fig. 1C–G, Movie EV2), as described previously (Colin-York et al, 2019a, 2019b; Kaizuka et al, 2007; Varma et al, 2006; Murugesan et al, 2016; Hashimoto-Tane et al, 2011). However, when a similar analysis of TCR movement was performed in Primary T cells, it revealed a significant deviation from the retrograde movement trajectories, where ~40% of motile TCRs showed a net outward ("anterograde") movement as well (Fig. 1C–G,

Movies EV3 and EV4), while average TCR speed was comparable in Primary (42.42 ± 1.75 nm/s) and Jurkat T cells (45.95 ± 2.91 nm/ s) (Fig. 1F). The speed of the TCR outward (38.37 ± 3.04 nm/s) and inward (45.51 ± 6.24 nm/s) fractions in Primary T cells was found to be comparable as well (Fig. 1G). To examine if the outward fraction of TCR constitutes free anti-CD3 antibodies on the bilayer that are expected to exhibit multidimensional diffusion on the bilayer, we plotted the TCR directional fractions against intensities, given that free-anti-CD3 monomers are likely to be of lower intensities. We found that the directional fraction of TCR was independent of intensity (Fig. 1H), indicating that TCR microclusters do contribute to both anterograde as well as retrograde fractions. TCR microclusters are known to form preferentially in pSMAC and dSMAC zones and then migrate towards cSMAC (Varma et al, 2006; Valitutti et al, 2010; Campi et al, 2005; Monks et al, 1998). To examine if the anterograde TCR movement originates in a specific subsynaptic zone, we performed TCR movement analysis within radially segmented sub-synaptic zones. The results showed a lack of bias for outward movement in any subsynaptic zone, indicating that the anterograde movement of TCR can originate anywhere between cSMAC and the cell periphery (Fig. 1I). Together, these data indicate that while the movement of engaged TCR is primarily centripetal in Jurkat cells, a significant fraction translocates outwards in Primary T cells —a behavior that can originate anywhere across the synaptic interface.

### Novel actin waves at synapse guide anterograde TCR microcluster migration

The centripetal TCR flow is known to be guided by a retrograde F-actin flow, which arises via actin polymerization and rearrangement following antigen recognition (Murugesan et al, 2016; Babich et al, 2012; Kumari et al, 2013; Kumari et al, 2012). We wondered if the anterograde movement of TCR that we observed in Primary T cells also relies on an outward actin flow, if any. First, to investigate whether the anterograde actin flows exist in the two T cell types, we imaged Jurkat T cell line stably expressing LifeAct-citrine (Colin-York et al, 2019a) or Primary CD8+ T cells derived from LifeAct-GFP expressing mice (Mandal et al, 2023), since LifeAct has previously been used as a reliable marker of actin dynamics in immune cells (Kumari et al, 2015; Reversat et al, 2020; Kumari et al, 2020; Brunetti et al, 2021; Gaertner et al, 2022; Riedl et al, 2008) (Fig. 2A, Movies EV5 and EV6). Indeed, we found the presence of an anterograde actin movement within the context of an overall retrograde flow at the synapse in Primary T cells (Fig. 2A, Movie EV6). Both the speed (41.29 ± 2.43 nm/s) as well as the directional fractions (0.45 ± 0.05) of actin (Fig. 2A,B) were comparable with those of TCR (Fig. 1E,F; 42.42 ± 1.75 nm/s and 0.43 ± 0.04, respectively). The actin anterograde flow was missing in Jurkat cells (Movie EV5). Thus, while the actin flow is primarily centripetal in Jurkat cells, directionally divergent actin flows exist in Primary T cells with speed and directions similar to those of TCR movement.

How could multidirectional flows, anterograde as well as retrograde, arise within the same subsynaptic locations? At the synapse, the retrograde actin flows are generated by actin polymerization at the plasma membrane in the dSMAC zone followed by contraction of resultant F-actin by Myosin II

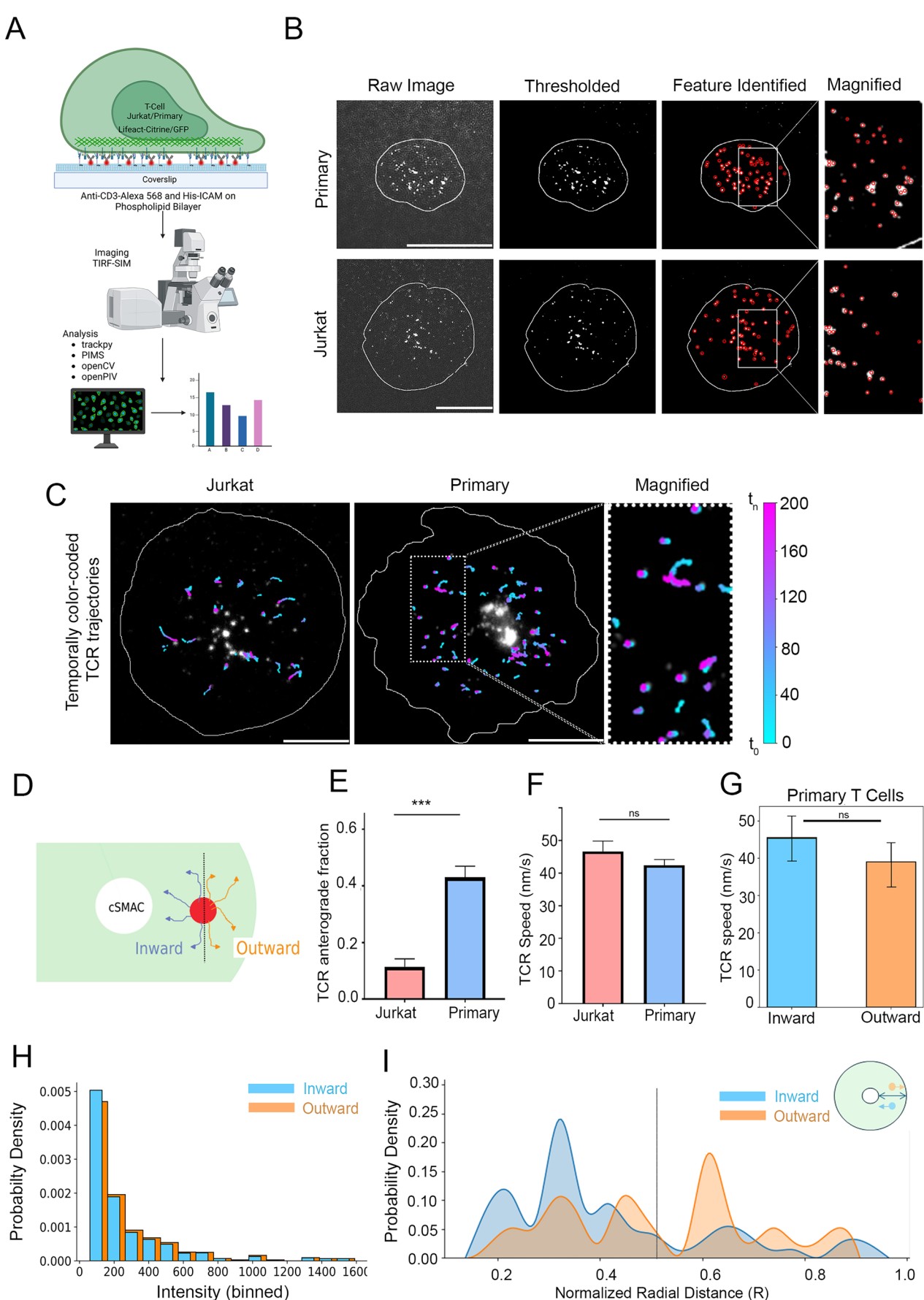

◄ **Figure 1. Live imaging of T cell immunological synapses using Total Internal Reflection-Structured Illumination Microscopy (TIRF-SIM) at super-resolution scale to examine TCR microcluster movement dynamics.**

(A) Schematic of the experimental setup utilized to visualize and analyze immunological synapses in live cells. The list shows the software employed in the analysis pipeline (see 'Methods'). (B) Examples of image processing steps underlying automated microcluster processing and detection in TIRF-SIM images. The white line outlines cell boundary, determined by overall actin distribution in the cell across the imaging duration. (C) Pseudocolored images showing trajectories of TCR clusters tracked within 200 s. The trajectories were color-coded based on their position in individual frames in reference to the initial frame. The reference color bar is shown on the right, where $t_0$ refers to the time of initiation of imaging ($t_0$ = synapse duration of <120 s), and $t_n$ refers to the time in the nth frame. (D–G) A schematic showing the determination of anterograde "inward" (retrograde) and "outward" (anterograde) motion of TCR (D), and quantitative estimates of the fraction of anterograde moving TCR (E) (Jurkat 0.11 ± 0.03; Primary 0.43 ± 0.04; data represented as Mean ± SEM), and speed (nm/s) and speed (F) (Jurkat 45.94 ± 2.91 nm/s; Primary 42.42 ± 1.75 nm/s; data represented as Mean ± SEM) of TCR ($n \geq 10$). The speed of outward vs inward moving TCR (nm/s) is shown in (G). The p-values of comparison between Jurkat and Primary cell data in (E), (F), and (G) are 0.00018, 0.418, and 0.536, respectively, obtained using Mann–Whitney non-parametric two-tailed test. (H) The directional fraction of TCR is independent of microcluster fluorescence signal intensities. The Y-axis represents the probability density of feature fluorescence intensities (Probability density = $\frac{frequency}{\sum of\ frequencies * bin\ width}$), and the X-axis presents intensity bins. Note that the inward and outward fractions show a similar distribution of intensities (p-value of the comparison between the inward and outward fractions is 0.107, obtained using the Mann–Whitney two-tailed non-parametric test). (I) Stratification of TCR tracks based on their position within radial zones of the synapse, where R = 0 refers to the cSMAC boundary, and R = 1 refers to the cell's outermost boundary. The Y axis represents the probability density of TCR initial positions within the radial zones described in D; p-value = 0.009 using Mann–Whitney non-parametric two-tailed test. Scale bar, 5 μm. Source data are available online for this figure.

(Murugesan et al, 2016; Babich et al, 2012, 2012; Fritzsche et al, 2017). To investigate the mechanism of anterograde flow generation at the Primary T cell synapse, we closely examined the average behavior of overall actin flows at the synapses. We found distinct and repetitive "wavefronts"-like structures arising and expanding outwards in the lamellar zone of the synapse. Such wave-like excitations of actin cytoskeleton, although not in the context of simultaneously opposite flows, have previously been observed in other cellular systems, and are referred to as "actin waves" (Riedl et al, 2008; Beta et al, 2023; Allard and Mogilner, 2013; Bretschneider et al, 2009; Stankevicins et al, 2020; Weiner et al, 2007; Riedl et al, 2023). To better demarcate actin wavefronts, we applied the Sobel edge detection algorithm (Mathur et al, 2016). For this, the raw images were first processed through a $5 \times 5$ pixel² ($208 \times 208$ nm²) Gaussian filter (Kumari et al, 2015). The algorithm then detected edges within a predetermined range of minimum and maximum intensity values within the gradients in the Gaussian filtered image (see 'Methods' for details). We then assessed the colocalization of TCR with the detected actin waves. If an edge coordinate (green lines in Fig. 2C) was present within eight nearest pixel neighbors (~60 nm radius) of the TCR (pseudocolored TCR tracks in Fig. 2C), the TCR was treated as colocalized with the actin wavefront for that time point. This analysis showed a high degree of colocalization between TCR and actin wavefronts (>70% average localization across the imaging window). This analysis was performed for entire image stream in a video, and the values of all TCR for all time points were averaged and presented in the graph (Fig. 3E). From this analysis we observed that the actin wavefronts were associated with TCR outward fraction (Fig. 2C, Movies EV7 and EV8). These observations implied that the net TCR microcluster movement at synapse may rely on vectorially opposite flows from retrograde flow towards cSMACs and actin waves expanding away from cSMAC (Fig. 2D). To gain further insights into this seemingly paradoxical observation, we explored the possibility of retrograde and anterograde flow co-existing and driving TCR microcluster movement within the synapse using a computational model.

Given that the movement of a fraction of TCRs was well-correlated with the actin wavefronts, we further investigated how the TCR may utilize actin dynamics for anterograde transport. There are two possible explanations to account for the anterograde migration of TCR and actin. First, the wavefronts of the

anterograde actin waves drive TCR movement outwards ("mode 1"), and second, the actin flows in the actin waves guide TCR outwards ("mode 2"). We interrogated these two possibilities using a mechanistic model of anterograde TCR microcluster movement.

We modeled the TCRs as tracer particles ("tracers" henceforth) with a default retrograde flow, and overlayed them on the movement of the experimentally detected actin wavefronts (mode 1; Fig. 2E, Movie EV9) or entire actin flow represented by the PIV flow field (mode 2; Fig. 2E, Movie EV10) of the cell. In the simulation, the TCR were allowed to transport in the direction and at the speed indicated by the nearest PIV arrow, provided that the velocity exceeded a given threshold value measured from experimental PIV data. The association of TCR with anterograde flow was first assessed by analyzing the TCR radial displacement values in the experimental data. The instances of TCR-anterograde movement (green trajectories, Fig. 2F left graph) or retrograde migration (gray trajectories, Fig. 2F left graph) were plotted. The trajectory values were binned to derive the probability distribution of time the microclusters spent in the anterograde and retrograde directions (Fig. 2F, right graph). For investigating the two possible aforementioned modes of TCR-actin anterograde flow association, we measured the outward time distributions of tracers from simulations and compared them to experimental values (Fig. 2G). We observed that if we allowed tracers to follow the actin wavefronts in our simulations, we got time distributions very close to those obtained from experimental values (Mode 1; red trace, Fig. 2G, left graph). Whereas, for mode 2, speed thresholds (PIV arrows that are less than a certain threshold) significantly influenced the accuracy of the fit. For mode 2, the speed thresholds spanning 13–33 nm/s resulted in the best match with experimental time distributions (Fig. 2G, right graph), thresholding below and above this range led to poorer match with experimental values. These findings suggest that for anterograde movements, the wavefronts drive TCR microcluster anterograde movement, while only actin flows with a specific range of speed could associate with TCRs. Although we do not have any evidence to interrogate the origin of the specific speed range, it is possible that it originates from the speed of the wavefronts themselves. The computational analysis also implied that TCR even if moving retrogradely, can rapidly switch the direction of their motion after encountering the expanding actin wavefronts.

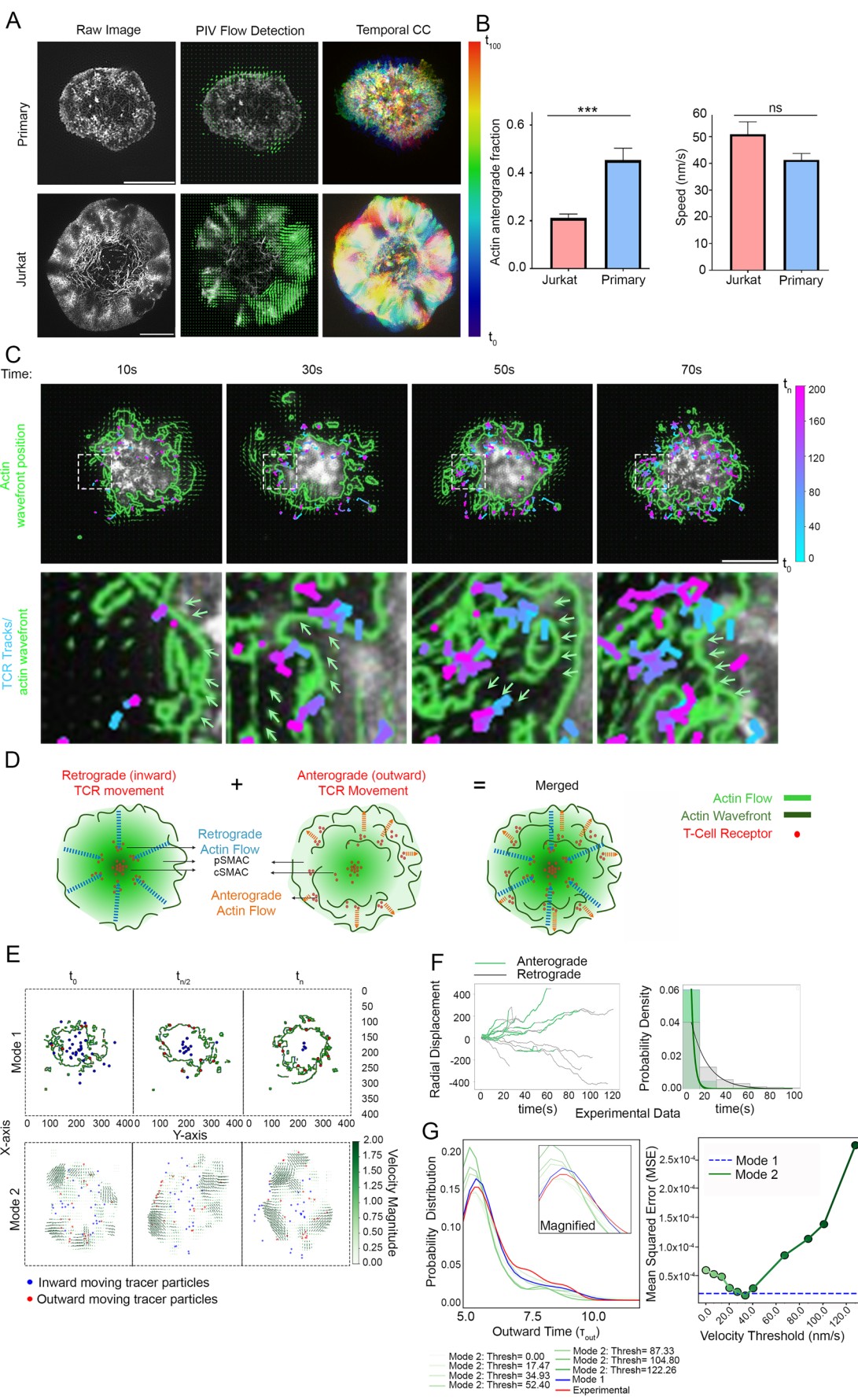

**Figure 2. Actin dynamics at the synapse, and a computational model of actin-TCR interaction at the synapse.**

(A) Particle Image Velocimetry (PIV) analysis of actin flows in Primary (top middle panel) and Jurkat T cells (bottom middle panel). The arrows in PIV images show actin flow directions at a representative time point during the duration of imaging. The rightmost panels show temporal color coding of actin flows in the cells shown on the left. The color bar shows the pseudo colors corresponding to the relative temporal positions of actin features in the duration of 100 s. Note that while the pseudocoloured actin flows are smooth in Jurkat (bottom right panel) indicating a persistent motion, pseudocolored actin flows in the Primary cell (top right panel) are heterogeneous indicating microscale directional fluctuations. Scale bar, 5 μm. (B) Quantification of anterograde fraction (Jurkat 0.21 ± 0.02; Primary 0.45 ± 0.05; data represented as Mean ± SEM) (left graph) and speed (Jurkat 50.89 ± 4.68 nm/s; Primary 41.29 ± 2.43 nm/s; Data represented as Mean ± SEM) (right graph) of actin features in Jurkat and Primary cells (n ≥ 11 cells in each case); p-value is 0.0005 and 0.148, respectively, obtained by Mann–Whitney non-parametric two-tailed test. (C) Temporal sequence of actin and TCR dynamics at the synapse, with actin wavefronts' positions are outlined with green lines (see 'Methods'), and motile TCR tracks pseudocoloured to denote time similar to Fig. 1C. The green arrows in insets at bottom indicate the direction of actin wave migration. The bar on the right panel shows blue for time t = 0 and t = 200 frames. For clarity, the whole cell boundary is not shown in the images. Scale bar, 5 μm. (D) A schematic showing the components of the computational model (actin waves, actin retrograde flow, and TCR at the synapse). The left panel highlights the retrograde movement of both actin and TCR, while the middle panel highlights their anterograde movement. The panel on the right shows both components overlaid, as seen in Primary T cells. (E) Snapshots taken from the mechanistic computational model exploring the relationship between anterograde actin wavefronts, actin flows, and TCR tracers (red and blue dots). The snapshots are taken from Movie EV9 (Top panel) and Movie EV10 (Bottom panel). The top panel highlights mode 1 where the actin wavefronts travel in anterograde direction and interact with TCR tracers; the bottom panel involves mode 2, where the tracers interact with the entire actin PIV flow by moving in the direction indicated by the nearest PIV arrow of considerable magnitude. The default direction of tracer migration was retrograde. (F) Radial displacement values of individual TCRs, obtained from experimental data, where the initial position of all TCRs is taken as zero. A positive trajectory signifies anterograde TCR movement (green), whereas a negative trajectory (gray) signifies retrograde TCR movement. The plot represents TCR tracking values from a representative single cell. This analysis was performed for 10 cells, and a similar trend was seen for all cells. The plot on the right in (F) shows the probability density of the periods in inward (Gray and outward (Green) motion across all primary cells. (G) Shows the outward time distributions of tracers' trajectories obtained from experiments (red trace), simulations of mode 1 (blue trace) and simulations of mode 2 (green traces) in the computational model. Varying shades of green in (G) represent different PIV threshold values during mode 2 simulation, the threshold values for each trace are shown at the bottom of the graph. The distributions were made continuous by sampling bin points using a Gaussian KDE and interpolating between the points. The graph on the right in (G) shows the mean squared error values of the traces from mode 1 and mode 2 simulations, when compared with experimental values. Source data are available online for this figure.

## WASP mediates coupling between anterograde actin waves and TCR microclusters

To address the hypothesis that a strong association (coupling') between TCR and actin anterograde flow exists and could drive the actin wave-dependent anterograde movement of TCR, we explored the molecular mechanisms mediating TCR-anterograde actin flow association. We first tested if activation of nucleation factor Arp2/3 complex can mediate the coupling, since it can associate with TCR microclusters on one hand and with F-actin as a nucleation factor on the other hand, using pharmacological inhibitor CK666 that prevents Arp2/3 complex activation in T cells (Kumari et al, 2015, 2020; Hetrick et al, 2013). Treatment of cells with 50 μM CK666 did not alter TCR outward fraction compared to the DMSO-treated control cells (Anterograde TCR fraction Primary T cell control 0.43 ± 0.04, CK666 treated 0.45 ± 0.03; TCR speed Primary T cell 42.42 ± 1.75 nm/s, CK666 treated 41.34 ± 4.11 nm/s; upper panels in Fig. 3A,B; Movies EV12 and EV13), and the overall TCR migration direction remained significantly different from the largely retrograde TCR movement in Jurkat cells (Movie EV11, red horizontal line in the Fig. 3B left graphs). Consistently, F-actin flow speed and directional fraction also remained unchanged in CK666-treated cells (Yang et al, 2012) (anterograde actin fraction Primary T cells 0.45 ± 0.05, CK666 treated 0.52 ± 0.02; actin speed Primary T cell 41.29 ± 2.43 nm/s, CK666 treated 38.43 ± 1.29 nm/s; lower graphs in Fig. 3B). These results indicated that branched actin nucleation via the Arp2/3 complex does not spontaneously guide the anterograde microclusters via anterograde flow. Next, we tested if the Wiskott-Aldrich Syndrome Protein (WASP) could enable the TCR-anterograde flow coupling since WASP can directly interact with TCR microcluster signalosome proteins (Sasahara et al, 2002; Ngoenkam et al, 2021; Cannon et al, 2001), as well as with F-actin (Marchand et al, 2001). We examined TCR motility dynamics in WASP-/- Primary CD8+ T cells expressing LifeAct-GFP. The WASP-/- cells showed significantly lower anterograde movement of

TCR when compared to control cells (Anterograde TCR fraction WASP-/- 0.19 ± 0.03, TCR speed 39.69 ± 3.54 nm/s) (lower panels in Fig. 3A, upper graphs in Fig. 3B, Movie EV14), and even though the anterograde flow and actin waves were still preserved in these cells (Anterograde actin fraction in WASP-/- 0.499 ± 0.02; actin speed 50.06 ± 3.04 nm/s; lower panels in Fig. 3B). Both CK666 treatment and WASP-/- cells also showed marginally lower cortical F-actin fluctuations in comparison with Primary control cells (kymographs in Fig. EV2A,B). Furthermore, the anterograde actin waves were not exclusively present in Primary CD8+ T cells, murine Primary CD4+ T cells exhibited similar actin flows and waves with comparable directions and speed (Fig. EV3A–D, Movies EV15 and EV16; actin anterograde fraction CD4+ = 0.45 ± 0.02, CD8+ = 0.47 ± 0.03; actin speed CD4+ = 51.54 ± 1.83 nm/s, CD8+ = 45.15 ± 4.63 nm/s) when imaged at high temporal resolution on SLBs using TIRF microscopy.

To further investigate the F-actin dynamics that could enable simultaneous anterograde and retrograde flows in Primary T cells, we treated the Primary CD8+ T cells with 1 μM Jasplakinolide which leads to F-actin stabilization and loss of dynamics, using conventional TIRF microscopy at high temporal resolution. The treatment did not alter actin anterograde fraction (0.39 ± 0.02; Fig. EV3A–C), although it did cause reduction in speed of actin flows (actin speed 23.73 ± 1.23 nm/s) (Fig. EV3D, Movie EV17). A converse treatment of cells with 200 nM Latrunculin, which inhibits actin polymerization by sequestering free actin monomers, did not alter actin anterograde fraction either (0.48 ± 0.01), and reduced actin flow speed (32.27 ± 1.32 nm/s; Fig. EV3A–D, Movie EV18) similar to Jasplakinolide. Further analysis of the actin directional fraction revealed that both Jasplakinolide and Latrunculin treatments reduced the number of both anterograde and retrograde actin features, resulting in overall unchanged directional fraction (Fig. EV3C). A similar analysis of actin flow speeds also revealed that drug treatments led to a reduction in both anterograde as well as retrograde speeds (Fig. EV3E). Thus, just as the previously characterized retrograde flows, the anterograde also rely on actin polymerization and

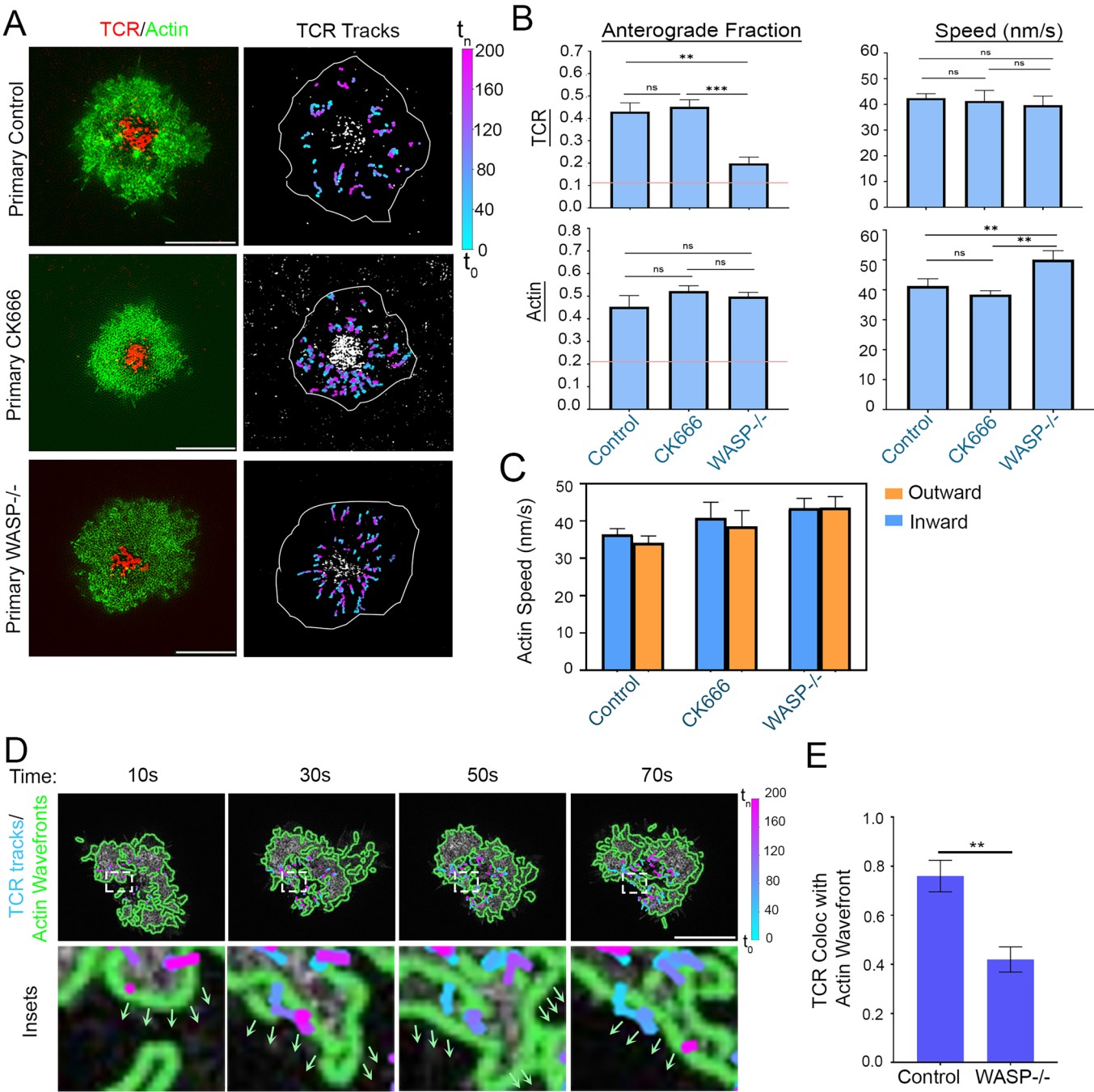

dynamics. Since a lack of WASP thus far was the only case where an uncoupling of anterograde waves and TCR movement was seen, we looked carefully at the TCR-actin wave coupling in WASP-/- cells. Analysis of actin wavefronts and TCR movement showed frequent "slip" events in WASP-/- cells, where growing wavefronts failed to associate with microcluster trajectories (Fig. 3D, Movies EV19 and EV20) and an overall lack of correlation between TCR movement and actin wavefronts (Fig. 3E). These results indicate that WASP or WASP-interacting proteins support TCR anterograde movement without playing a direct role in generating the anterograde flow of actin.

Using an in vitro SLB-based ligand reconstitution system that has previously been extensively utilized to study synapse signaling,

in combination with Primary T cells, our results reveal three novel aspects of antigen receptor and cytoskeletal behavior at the immunological synapse: First—there is an outward movement of TCR as well as F-actin at the synapse, second—the generation of outward traveling actin waves that form simultaneously with the inward flow of actin, a phenomenon that could explain the simultaneous anterograde as well as retrograde microcluster movement, and third—a key role of WASP in connecting receptor microclusters with actin waves for anterograde movement. The traveling actin waves have been described in a few cellular systems previously (Katsuno et al, 2015; Allard and Mogilner, 2013; Huang et al, 2020; Stankevicins et al, 2020; Wu et al, 2013), the unique

**Figure 3. Molecular determinants of TCR-anterograde flow coupling at the immunological synapse.**

(A) Synapses of LifeAct expressing Primary T cells with CK666 treatment and WASP-/- on SLBs containing anti-CD3-Alexa568 and Human ICAM-1 were imaged using TIRF-SIM. The panels on the left show the final snapshot from the videos showing gross TCR and actin distribution, while the panels on the right show tracks of TCR clusters throughout the videos, the cell boundaries demarcated by white borders. Scale bar, 5 μm. (B) Quantification of TCR motility (top graphs) and actin flows (bottom graphs) showing directional fraction (left; Anterograde TCR fraction—Primary T cell control 0.43 ± 0.04, CK666 treated 0.45 ± 0.03, WASP-/- 0.19 ± 0.03; Anterograde Actin Fraction- Primary T cells 0.45 ± 0.05, CK666 treated 0.52 ± 0.02; WASP-/- 0.49 ± 0.02) and speed in nm/s (right; TCR speed—Primary T cell 42.42 ± 1.75 nm/s, CK666 treated 41.34 ± 4.11 nm/s, WASP-/- 39.69 ± 3.54 nm/s; Actin Speed- Primary T cell 41.29 ± 2.43 nm/s, CK666 treated 38.43 ± 1.29 nm/s, WASP-/- 50.06 ± 3.04 nm/s). At least 10 cells were analyzed in each case, and the p-values of the comparison are shown in Table EV1, as obtained using the Mann–Whitney non-parametric two-tailed test; data represented as Mean ± SEM. (C) The speed of outward and inward moving actin features, represented from data in (B) (bottom right graph) (Data represented as Mean ± SEM). Average cell boundaries across the timelapse are represented by white outlines. (D) Tracking of actin waves (outlined in green lines) and TCR tracks (marked as color-coded trajectories) in LifeAct-GFP expressing WASP-/- Primary T cell synapses. The wave-like behavior of actin was visibly observed in WASP-/- T cells 80% of the time ($n = 15$). The green arrows show the direction of actin wave migration. (E) Determination of TCR-wavefront colocalization in WT or WASP-/- Primary T cells, as described in the 'Methods' section. For the analysis, a total of $n \geq 7$ cells were used, and the p-value of comparison is 0.0012, as determined using the Mann–Whitney two-tailed non-parametric test (Data represented as Mean ± SEM). Source data are available online for this figure.

feature in lymphocyte synapse is that here they exist simultaneously, in space and time, with a persistent retrograde actin flow. Whether the generation and sustenance of the two vectorially opposite actin flows at the same subcellular location follows cytoskeletal design principles unique to immune cells, and their underlying molecular ramifications that enable this highly counter-intuitive and specialized cytoskeletal dynamics, will be a subject of future research.

Surprisingly, in the Jurkat cell line, which has been a system of choice to investigate T cell activation, the majority of microclusters move only centripetally towards cSMAC. While the reason for the centripetal TCR migration bias is likely to be the absence of actin waves in Jurkat cells, why the actin waves fail to form in Jurkat cells is not clear at this stage. It is possible that some of Jurkat-specific genomic variations underlie the difference (Gioia et al, 2018). A lack of actin waves and TCR anterograde fraction in Jurkat cells—the most widely used cellular system for T cell biology—also explains at least to some extent why the anterograde microcluster movement has not been observed previously. In addition, the higher spatiotemporal resolution afforded by TIRF-SIM enabled the identification of fast microscale F-actin dynamics at early T cell synapse combined with unbiased cluster tracking, which could potentially be missed by routine cluster imaging and tracking methodologies. The actin waves described here can now be added to the microscale actin organizations at the immunological synapse that regulate the signaling architecture of T cells during activation (Murugesan et al, 2016; Kumari et al, 2015; Carisey et al, 2018; Fritzsche et al, 2017).

An interesting and somewhat confounding result we obtained was the selective use of WASP (or WASP-interacting proteins) for coupling microclusters only to anterograde flow and not to the retrograde flow. The generation and sustenance of retrograde actin flows that guide the retrograde movement of clusters is known to be mediated by the formation of actin arcs and their contractions (Murugesan et al, 2016; Kaizuka et al, 2007; Babich et al, 2012). We found that while the retrograde flow is still the major contributor to the cluster migration, it seems to be coupled to the microcluster independently of WASP. The identity of the molecules that couple TCR to the actin arcs and retrograde flows in general, whether WASP directly couples antero-grade flows to TCR microclusters or is associated with its interacting proteins, and the physical principles sustaining the coupling in both anterograde as well as retrograde movement remains to be investigated. Our simulations imply a speed selectivity in association of TCR with actin flows for anterograde movement. Whether and how the

permissive actin speeds are coordinated with WASP-dependent molecular mechanism for coupling TCR with anterograde flow will also be interesting to explore in the future.

Although we did not investigate the immunological significance of the anterograde TCR microcluster movement directly, several elegant studies have demonstrated the significance of TCR movement in the synaptic context (Saliba et al, 2019; Choudhuri et al, 2014; Sasahara et al, 2002) including its role in antigen receptor signal desensitization, degradation, exocytosis, and intercellular communication. The anterograde movement may alter or at least delay a sizeable fraction of TCR from these eventualities. Another tempting possibility is that the anterograde co-movement of microclusters along with actin waves prolongs their signaling lifetime, and may even augment signal amplification via long-range cytoskeletal consolidation (Pielak et al, 2017; Pathni et al, 2022). Similarly, the sustained generation of actin waves and subsequent lamellar extensions may increase the cell surface area of synapse for better coverage of antigens on the opposing interface for further TCR-MHCp engagement events (Hui et al, 2015; Bunnell et al, 2002; Barda-Saad et al, 2005; Yi et al, 2019; Wahl et al, 2019). Finally, the propagation of lamellar waves will also likely impose mechanical forces on pre-existing as well as newly formed ligand-receptor bonds, including the TCR-MHCp pairs. Whether and how all of the above exciting possibilities combine at the synaptic interface to impact overall T cell antigen sensitivity, affinity discrimination, and signaling lifetime—essential features of early lymphocyte activation that rely on antigen receptor dynamics—will be exciting to explore in the future.

## Methods

**Reagents and tools table**

| Reagent/Resource | Reference or Source | Identifier or Catalog Number |
|---|---|---|
| **Experimental models** | | |
| C57BL/6_LifeAct_GFP | Colin-York and Kumari et al. J Cell Sci (2020) 133 (5): jcs232322. | N/A |
| C57BL/6 _LifeAct_GFP_WASP-/- | Colin-York and Kumari et al. J Cell Sci (2020) 133 (5): jcs232322. | N/A |

| Reagent/Resource | Reference or Source | Identifier or Catalog Number |
|---|---|---|
| Jurkat 1G4 Cells | Lee et al. Sci Rep 7, 5416 (2017). | N/A |
| **Recombinant DNA** | | |
| LifeAct-mCitrine | Addgene | Catalog# 54733 |
| LifeAct-EGFP | Addgene | Catalog# 58470 |
| **Antibodies** | | |
| Anti-Mouse CD3ε (Clone 145- 2C11) | Biolegend | Catalog# 100302 |
| Anti-Human CD3 (Clone OKT3) | Biolegend | Catalog# 317302 |
| **Oligonucleotides and other sequence-based reagents** | | |
| | N/A | |
| **Chemicals, enzymes and other reagents** | | |
| RPMI-1640 | GIBCO | Catalog # 11875093 |
| Fetal Bovine Serum | GIBCO | Catalog # A5256701 |
| L-glutamine | Thermo Fisher | Catalog #A2916801 |
| Sodium Pyruvate | Thermo Fisher | Catalog # 11360070 |
| Penicillin/Streptomycin | Thermo Fisher | Catalog # 15140122 |
| PMA/Ionomycin | eBioscience™ Cell Stimulation Cocktail 500X | Catalog # 00-4970-93 |
| TheraPEAK® X-VIVO® 10 Serum-free Hematopoietic Cell Medium | Lonza | Catalog #: BEBP02-055Q |
| Human IL-2 Recombinant Protein | PeproTech | Catalog # 200-02-50UG |
| HEPES | Gbioscience | Catalog #:RC-060 |
| 6X His-ICAM1 | Thermo Fisher | Catalog #10346H03H50 |
| NaCl | Merck(Emparta) | CAS #7647-14-5 |
| KCl | SDFCL | CAS #7447-40-7 |
| Glucose | Sigma-Aldrich | CAS #50-99-7 |
| $CaCl_2$ | SDFCL | CAS #10043-52-4 |
| $MgCl_2$ | Sigma-Aldrich | CAS #7786-30-3 |
| $NiCl_2$ | Qualigens | CAS #7791-20-0 |
| Streptavidin | Thermo Fisher | Catalog # 434301 |
| DOPC | Avanti Lipids (Sigma-Aldrich) | Cat# 850375 |
| NTA-DGS | Avanti Lipids (Sigma-Aldrich) | Cat# 790404 |
| Cap-Biotin | Avanti Lipids (Sigma-Aldrich) | Cat# 870273 |

| Reagent/Resource | Reference or Source | Identifier or Catalog Number |
|---|---|---|
| CD8 T-cell isolation kit | Stem Cell Technologies | Catalog # 19853 |
| CK666 | Merck | CAS # 442633-00-3 |
| Jasplakinolide | Calbiochem | CAS # 102396-24-7 |
| Latrunculin | Calbiochem | CAS# 76343-93-6 |
| **Software** | | |
| MATLAB | https://www.mathworks.com/products/matlab.html | N/A |
| GraphPad Prism8 | https://www.graphpad.com/features | N/A |
| Adobe Photoshop | Adobe | N/A |
| Image J | https://imagej.net/ij/ | N/A |
| Trackpy | https://pypi.org/project/trackpy/ | N/A |
| OpenCV | https://opencv.org/blog/edge-detection-using-opencv/ | N/A |
| OpenPIV | https://openpiv.readthedocs.io/en/latest/ | N/A |
| **Other** | | |
| 25 mm round #1.5 glass coverslips | VWR | Catalog# MARI0117650 |

## Cell isolation, culture, activation, and pharmacological treatments

Mouse CD8+ T cells were isolated either from wild-type C57BL/6 mice, or from wild-type C57BL/6 and WASP-/- C57BL/6 mice expressing LifeAct-GFP, using the CD8+ T Cell Isolation Kit (Stem Cell Technologies) following the manufacturer's protocol. The cells were cultured in sterile RPMI-1640 medium (GIBCO), supplemented with 10% fetal bovine serum (GIBCO), 2 mM L-glutamine (Thermo), 1 mM sodium pyruvate (Thermo), and 1% Penicillin/Streptomycin solution (Thermo). Post isolation, the cells were activated using PMA/Ionomycin (eBioscience™ Cell Stimulation Cocktail 500X), since activation using anti-CD3/CD28 is severely compromised in WASP-deficient CD8+ T cells (Mandal et al, 2023). The cells were maintained at a density of $0.5–2 \times 10^6$ cells/ml in a 37 °C incubator with 5% $CO_2$ and humidity. The Jurkat T cells utilized in this study were the 1G4 CD8+ T cell receptor variant, where the original endogenous T cell receptor was replaced human NY-ESO-specific CD8+ T cell receptor (Lee et al, 2017). The 1G4 Jurkat cells were lentivirally transduced to generate stable cell lines expressing LifeAct-citrine (Colin-York et al, 2019b), and maintained in RPMI culture media mentioned above. At the time of imaging, the culture media was replaced with imaging media (TheraPEAK® X-VIVO® 10 Serum-free Hematopoietic Cell Medium (Lonza) supplemented with IL2 (10 units/ml), Fetal Bovine Serum (10%), and HEPES (10 mM), and live imaging was performed in imaging media.

## Glass-supported lipid bilayers reconstituted with ligands

The supported lipid bilayers (SLBs) were prepared as described previously (Crites et al, 2015). Briefly, the bilayers were deposited on 25 mm round #1.5 glass coverslips (VWR), which were cleaned with piranha solution (sulfuric acid: hydrogen peroxide, 3:2 ratio) for 30 min, washed extensively with MilliQ water, and allowed to dry with ambient air at room temperature. The coverslips were then coated with a 5 µl solution of equal volumes of DOPC liposomes (0.4 mM) and liposomes containing 12.5 mol% $Ni2+$-NTA-DGS (0.4 mM), and 0.05 mol% cap biotin phosphatidylethanolamine (0.4 mM), and 37.5 mol% DOPC (0.4 mM). The lipid film was then hydrated with HEPES Buffered Saline (20 mM HEPES, 140 mM NaCl, 5 mM KCl, 6 mM glucose, 1 mM $CaCl_2$, 2 mM $MgCl_2$; 'HBS') supplemented with 1% human serum albumin (HSA), pH 7.2, and washed several times with HBS. SLBs were then incubated with 5% bovine serum albumin supplemented with 100 µM $NiCl_2$ and 1 µg/ml streptavidin (to couple to biotin sites) for 30 min. SLBs were then washed extensively with HBS and incubated with mono-biotinylated 2.5 µg/ml hamster anti-CD3ε (2C11 for murine cells and OKT3 for Jurkat cells) and 1 µg/ml 6X His-ICAM1 to activate T cells. The mono-biotinylation of antibodies was assessed prior to the use using flow cytometric calibration of ligand binding on glass bead supported lipid bilayers (Bangs Lab) (Choudhuri et al, 2014; Fleire and Batista, 2009), and mobility of planar glass -supported bilayers was assessed prior to incubation with cells by examining the recovery profiles of Alexa568-anti-CD3 in photobleached zones on the lipid bilayers, and the recovery was found to be greater than 80% within 1 min of photobleaching in all experiments, consistent with high mobility of bilayers. Typically, only one cell/bilayer was imaged, since in the duration of imaging synapses matured to late stage in other cells on the same bilayer, and were not suitable for studying flows at early synapse.

## Live-cell super-resolution extended TIRF-SIM

Extended total internal reflection fluorescence structured illumination microscopy (TIRF-SIM) was conducted using a 488-nm laser (Coherent, SAPPHIRE 488–500), which was directed through an acousto-optic tunable filter (AOTF, AA Quanta Tech, AOTFnC-400.650-TN). The laser beam was expanded and directed into a phase-only modulator composed of a polarization beam splitter, an achromatic half-wave plate (Bolder Vision Optik, BVO AHWP3), and a ferroelectric spatial light modulator (SLM; Forth Dimension Displays, SXGA-3DM). The diffraction pattern produced by the grating on the SLM passed through a polarization rotator, consisting of a liquid crystal variable retarder (LC, Meadowlark, SWIFT) and an achromatic quarter-wave plate (Bolder Vision Optik, BVO AQWP3). This configuration rotated the linear polarization of the diffracted light to preserve s-polarization, optimizing pattern contrast across all orientations.

To isolate the ±1 diffraction order, a hollow barrel mask driven by a galvanometer optical scanner (Model 623OH, Cambridge Technologies, Bedford, MA) was used to block higher-order diffraction light. The selected beams were then focused onto the back focal plane of a high-NA objective (Olympus Plan-Apochromat × 100 Oil-HI 1.57NA) as two spots on opposite sides of the pupil. After collimation by the objective, the two beams interfered at the coverslip-sample interface at an angle exceeding the critical angle for total internal reflection, generating an evanescent standing wave of excitation that extended approximately 100 nm into the sample axially, with a laterally modulated sinusoidal pattern. This modulated pattern was a low-pass filtered, demagnified image of the grating displayed on the SLM. The emitted fluorescence was collected by the same objective, separated from the excitation light using a dichroic mirror, and imaged onto a sCMOS camera (Hamamatsu, Orca Flash 4.0 v2 sCMOS), where the structured fluorescence raw data were recorded. Cell samples were imaged in a micro-incubator (H301, Okolab, Naples, Italy) under physiological conditions (37 °C and 5% $CO_2$). For each time point, three raw images were acquired at successive phase steps (0, 1/3, and 2/3) over the sinusoidal illumination pattern. This process was repeated with the excitation pattern rotated by +120° or −120° relative to the initial orientation. Phase stepping and pattern rotation were controlled by translating and rotating the grating image on the SLM. A total of nine raw images were acquired for each excitation wavelength before moving to the next. This acquisition cycle was repeated for each time point. Finally, the raw images were processed and reconstructed into SIM images. TIRF-SIM data were collected from at least 50 individual bilayers (technican replicates) across ≥ three biological replicates.

## Primary T cell nucleofection

The isolated murine CD8+ T cells were activated using 25 ng/ml Phorbol 12-myristate 13-acetate and 0.5 µM Ionomycin in RPMI-1640 medium (GIBCO), supplemented with 10% fetal bovine serum (GIBCO), 1 mM sodium pyruvate (Thermo), and 1% Penicillin/Streptomycin solution (Thermo) overnight. The activated cells were then replenished with fresh media supplemented with 10 units/ml of IL-2. 48–72 h post activation $5 \times 10^6$ cells were resuspended in nucleofection buffer with 5 µg of LifeAct-EGFP plasmid construct and subjected to nucleofection with Amaxa Nucleofector II (Lonza) using the X-001 protocol. The cells were immediately transferred to pre-warmed media (37 °C), and the experiment was performed within 24 h.

## Pharmacological Inhibition and Live Imaging using conventional TIRF microscopy

On Nikon Ti2 TIRF microscope, the TIRF alignment was done on 100X oil-objective and 1.5X projection lens with 488 nm laser at the focal plane. The live imaging stage was humidified and set at 37 °C. The LifeAct-GFP expressing cells were added to the SLB in 18 well chambers at a density of $10^5$ cells/well and imaged soon after. Individual cells were imaged at 10% laser power, 200 ms exposure at an interval of 500 ms for 5 min. For wells with pharmacological inhibition, LifeAct-EGFP expressing CD8+ T cells were first briefly allowed to attach and spread on SLBs briefly, and 1 µM of Jasplakinolide or 200 nM of Latrunculin was added to the wells immediately after, and cells were then imaged under above mentioned conditions.

## Data plotting and statistics

Prior to analysis, the timelapse data was anonymized by removing the labels. Data then obtained using image analysis methodologies described in the aforementioned sections was plotted using either

MATLAB or Prism (GraphPad). Cartoons and schematics were generated using BioRender. Figures were assembled using Adobe Photoshop (Adobe). Statistical comparison between groups was carried out using Mann–Whitney two-tailed non-parametric distribution. The number of data points within distributions is mentioned in the figure legends and correspond to average values obtained from cells in one experiment. Unless otherwise mentioned each bar in graphs represents Mean ± SEM values across the indicated number of cells in corresponding figure legends. 'n' values in the Figure legends represent number of cells from the experiment (technical replicates, where each T cell was imaged on one individual bilayer). Unless otherwise mentioned, experiments were performed ≥3 times (biological replicates where cells were obtained from different animals).

## Tracking analysis

Images were analyzed using Python image processing packages trackpy (Crocker and Grier, 1996), openCV (Cespedes et al, 2022), and openPIV (Gida et al, 2020). The videos are first processed using the ImageJ program: the images were converted to 8-bit grayscale image files, after which the images were filtered by applying a Gaussian filter of radius of 3 pixels (a window of radius 3 pixels was manually decided as it gave the best TCR detection when manually verified by colocalization with raw images to detect maximum clusters while avoiding artifacts, when tested for 6 technical replicates; Fig. EV1A). The images were then thresholded to identify individual microclusters as "particles" for tracking. To track TCR particle trajectories over time, we used dynamic tracking (Crocker and Grier, 1996). The probability for every new position of the particle was tracked using:

$$P(t_1, t_0, \vec{x}(t_0)) = \vec{x}(t_0) + \frac{\vec{x}(t_0) - \vec{x}(t_{-1})}{t_0 - t_{-1}}(t_1 - t_0)$$

Where, $P(t_1, t_0, \vec{x}(t_0))$ is the probability distribution of the particle at a time $t_1$ given the particle was at $\vec{x}(t_0)$ at time $t = t_0$. Calculation of the probability requires calculating the instantaneous velocity, given by: $\vec{v}(t_0) = \frac{\vec{x}(t_0) - \vec{x}(t_{-1})}{t_0 - t_{-1}}$. Here $\vec{x}(t_{-1})$ is the position of the cluster at a time previous to the current frame.

To ensure we track TCR microclusters and not any incidental free-floating debris on the SLB, we filtered out TCR trajectories with an end-to-end displacement less than 3.5 pixels (146 nm; ~5% of the radius of the T Cell). We then examined the directionality of the detected TCRs. Finally, to ensure that the detected TCR features were in the dSMAC and pSMAC areas of active TCR transport, and not in cSMAC, where a large fraction of TCR accumulates, we utilized actin images as a positive mask for the tracked area, since the actin cytoskeleton is depleted in the cSMAC zone (Kaizuka et al, 2007; Ritter et al, 2015). To ensure we correctly tracked TCR clusters in the synapse, calculated trajectories were manually verified using ImageJ software (Fig. EV1B). Both methods yielded comparable values for the speeds of TCR clusters which were further consistent with previously published values (Colin-York et al, 2019b). From the trajectory data, we could calculate the velocities and the directions of motile TCRs.

To quantify the fraction of inward and outward-moving TCR, we calculated the radial displacements of TCR from the cell center to the initial and final points in the trajectory. If the initial radial displacement was smaller than the final radial displacement, the trajectory was classified as an 'anterograde' trajectory and vice versa. We also extracted spatial data for positional distributions of anterograde and retrograde TCR fractions shown in Fig. 1I. To better visualize the distribution, we normalized the radial distances by the distance from the edge of the cSMAC to the cell periphery in all cells, such that the cell radius lay between 0 and 1 representing the periphery of the cSMAC and the cell periphery, in all cells. This allowed us to pool TCR trajectories across cells.

For tracking actin features in LifeAct images, the actin features were analyzed to generate flow data. We used Particle Imaging Velocimetry (PIV) flow algorithms using the openPIV library in Python to quantify LifeAct flow directions and speed. To validate the PIV tracking algorithm, we tested the results using two methods. First, we checked whether the algorithm gave retrograde actin velocities consistent with previous studies (Colin-York et al, 2019b). Second, we performed particle-based tracking of actin instead of flow-based analysis, as done for TCR clusters mentioned above. The actin images were processed using a Gaussian filter as described for TCR images, and again thresholded to pick out the brightest actin features. The directional fraction values from this method resulted in an average outward fraction of 0.44 as compared to 0.38 as done by PIV tracking.

Edge detection of actin wavefronts was performed using a Sobel edge detection algorithm (Mathur et al, 2016) to detect actin-rich wavefronts. First, the image was processed through a $5 \times 5$ pixel$^2$ ($208 \times 208$ nm$^2$) Gaussian filter (Kumari et al, 2015), which blurs the image so that only the most prominent edges are detected. To determine the appropriate edge gradients, we manually examined the pixel values of edge gradients of the wave features and took the average of intensities from 4 cells in each Primary control and Primary WASP-/- cells. From this measurement, we found the minimum edge gradient as 10 (arbitrary units, AU) and the maximum as 40 AU, which were then applied for image processing.

## Computational modelling

We developed a phenomenological model of TCR transport via the actin cytoskeleton. It has already been shown in the Jurkat cells that TCR couples to the actin retrograde flow and moves persistently toward the cSMAC region of the immunological synapse. In the mouse Primary cells, we found that the TCR movement correlates with both retrograde flow and anterograde flow. To model the anterograde movement of the Primary T cells, we assumed that the TCRs are featureless particles ("tracers") that are confined to move on the plane of the synapse. In this model, the tracers are constitutively coupled to the retrograde flow, so that they naturally move towards the cell center.

To gain insights into tracer-anterograde flow association, we investigated two possible modes of TCR transport by the actin cytoskeleton. We compared the two modes by measuring the outward and inward time distribution (the probability distribution of the amount of time the particle spends moving outward and inward, respectively, during its trajectory). We compared these outward time distributions from simulations to those obtained from experimental trajectories of TCR (1124 individual trajectories pooled from 10 cells; Fig. 2F).

Mode 1: We extracted the detected actin wavefronts from the experimental data from Fig. 2C (simulation space shown in

Fig. 2E). We then initialized 100 computational tracer particles distributed randomly across the synaptic interface. We gave the tracer particles a default retrograde motion with speed 50 nm/s (based on the speed of TCR in Jurkat); Mode 2: We extracted the PIV flow fields instead of wavefronts from the experimental data of Fig. 2A. We then initialized 100 tracer particles and gave them the default retrograde motion as done in mode 1. The tracer particles scanned a radius of 300 nm at each step and then associated with either the wavefront (mode 1) or average PIV features within scanning area of the tracer in each step (mode 2). Upon successfully encountering a wavefront or PIV flows, the tracer followed the expanding wavefront (mode 1) or the average velocity of the local PIV. We estimated the time distributions of tracer's trajectories and compared them with the experimental values. The graphs show data from all simulations, after using Gaussian KDE to determine the bin widths and then interpolating over the heights of the normalized histogram bins (Blumenthal and Burkhardt 2020; Cespedes et al, 2022; Kumari et al, 2013; Kumari et al, 2012).

## Data availability

Modeling codes have been uploaded to Zenodo: https://zenodo.org/records/17778174?token=eyJhbGciOiJIUzUxMiJ9.eyJpZCI6ImQzOWU1OGJiLTE5YTEtNDI5My1hYzIwLTcxNzVjVjNWE1M2UxOSIsImRhdGEiOnt9LCJyYW5kb20iOiI3ZkyNTUxMGExNjZlM2MzMDRiYzIxYTEwMjgwNWFkOCJ9.r05gEClMnPhQ6Yr8E3GTYFZfVKBwsl1Sq1MjZ70y1UsPOmF7kTS_irEzx5QjUi8dAoyT0yTKG0TCHoWVyYBx9g.

The source data of this paper are collected in the following database record: biostudies:S-SCDT-10_1038-S44319-025-00676-2.

## Peer review information

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

## Acknowledgements

We thank the Biology Divisional Microscopy facility and Central animal facility at the Indian Institute of Science; the Advanced Imaging Center at Janelia Campus, and J. Haddleston for help with the lattice light-sheet microscopy (The Advanced Imaging Center is jointly funded by the Howard Hughes Medical Institute and the Gordon and Betty Moore Foundation). SS acknowledges generous support from Axis Bank Center for Maths and Computing (OD/ACMC-23-0013) and SERB-DST India (SRG/2022/000163). Both SS and SK acknowledge generous support from the MoE-STARS grant. SK acknowledges Ashesh Dhawale, Kheya Sengupta and Pierre-Henri Puech for valuable analysis suggestions during early phase of the study; Prime Minister's Research Fellowship (Graduate fellowship) to SM; ANRF/SERB grant (SPG/2021/004030), an Infosys Young Investigator fellowship, and an Intermediate Fellowship from India Alliance DBT-Wellcome Trust (IA/I/23/1/506757).

## Author contributions

**Aheria Dey**: Data curation; Visualization; Writing—review and editing. **Samuel Z Khiangte**: Formal analysis; Methodology; Writing—review and editing. **Srishti Mandal**: Data curation; Validation; Visualization. **Huw Colin-York**: Data curation; Investigation. **Marco Fritzsche**: Resources; Methodology. **Sumantra Sarkar**: Formal analysis; Methodology; Writing—review and editing. **Sudha Kumari**: Conceptualization; Resources; Data curation; Supervision; Funding acquisition; Visualization; Methodology; Writing—original draft; Writing—review and editing.

Source data underlying figure panels in this paper may have individual authorship assigned. Where available, figure panel/source data authorship is listed in the following database record: biostudies:S-SCDT-10_1038-S44319-025-00676-2.

## Disclosure and competing interests statement

The authors declare no competing interests.

# Expanded View Figures

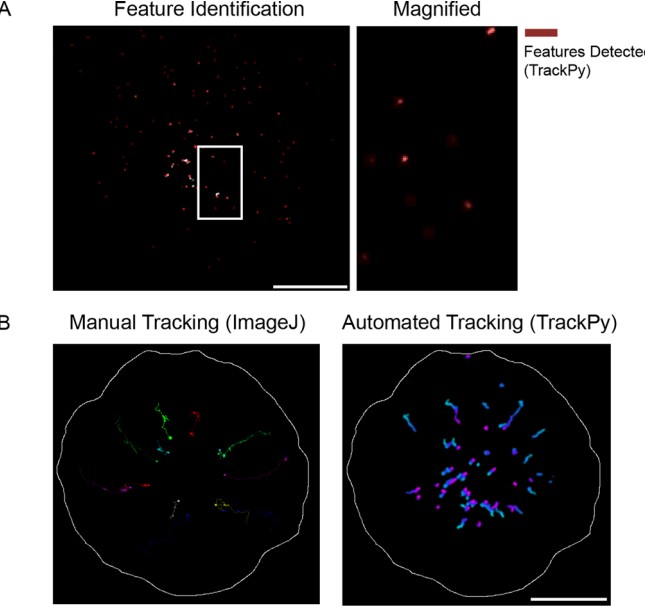

**Figure EV1.** **Validation of automated microcluster detection and tracking methodology.**

(**A**) Validation of automated TCR microcluster ("feature") detection. The position of the detected clusters are denoted in red pixels, the colocalization of the red pixels with the highest intensity pixels of the raw image (>150) was then measured and found to be 97.2%. (**B**) Validation of tracking routine. Manual tracking of TCR was performed in ImageJ where a particle trajectory was plotted by manually identifying the position of the particle at each time and then combining the identified points to form a track, the cell boundaries demarcated by white borders. The corresponding speeds of TCRs were compared and found to show similar results (manual: 45.72 ± 6.82 nm/s; automated: 48.26 ± 8.91 nm/s). Scale Bar = 5 μm. Source data are available online for this figure.

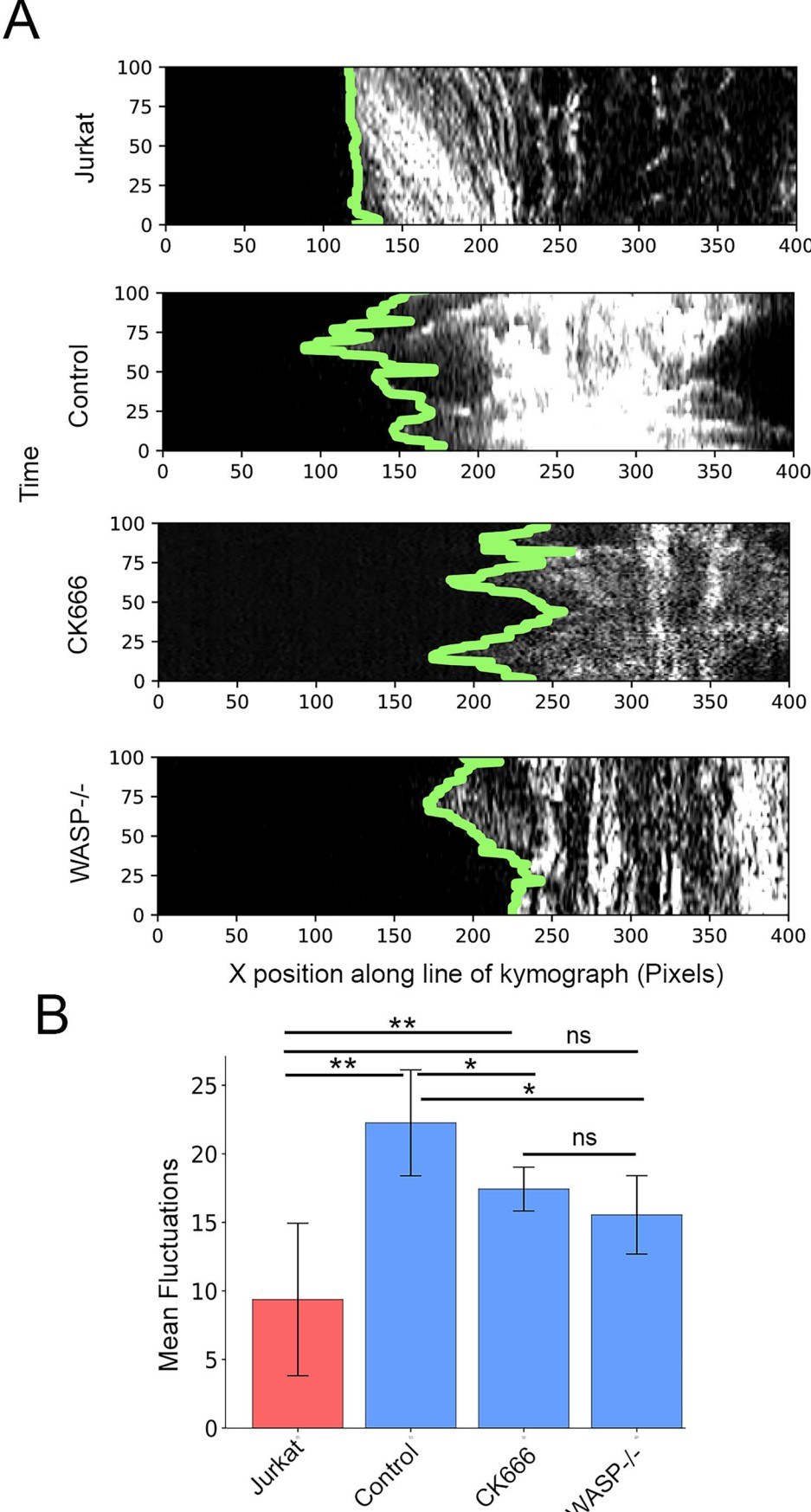

**Figure EV2. Comparison of cortical lamellar waves and fluctuations at synapse.**

(A) Kymographs showing actin waves in Jurkat (top panel) and Primary T cells (bottom three panels), the green lines indicate the evolution of actin boundary over time. (B) A comparison of the fluctuation from the mean of the cell edges (No. of cells analyzed $n \geq 5$) (RMS fluctuations $p$-values: Jurkat vs. Primary control- 0.001; Jurkat vs. Primary CK666- 0.006; Jurkat vs. Primary WASP-/- 0.072; Primary control vs. Primary CK666- 0.018; Primary control vs. Primary WASP-/- 0.017; Primary CK666 vs. Primary WASP-/- 0.214) (Data represented as Mean ± SEM). Source data are available online for this figure.

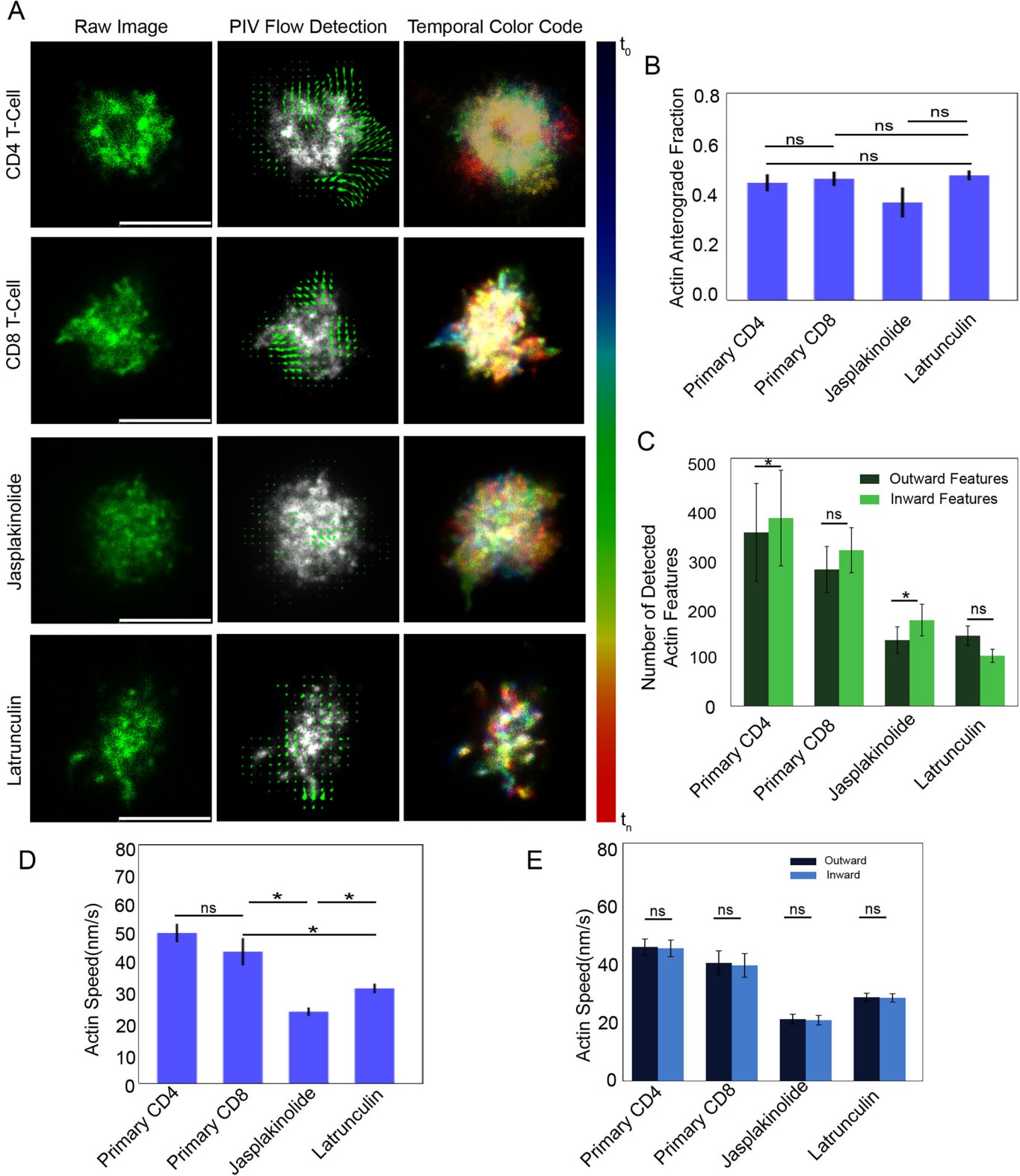

◀ **Figure EV3.   Pharmacological interrogation of actin polymerization dynamics in generation of anterograde and retrograde flows.**

(A) Snapshots of control or actin inhibitor treated T cells' synapses. The LifeAct distribution in mouse Primary CD4+ cells, and CD8 + T cells with and without treatment with 1 μM Jasplakinolide or 200 nM Latrunculin are shown in left panels. The middle panel shows the particle image velocimetry (PIV) images, and the right panel shows the temporal colour-coded images. Scale bar, 5 μm. (B) The ratio of the actin anterograde features are depicted in the graphs. The fraction of outward moving actin features are similar for CD4 and CD8 T cells. Treatment with Latrunculin A and Jasplakinolide does not significantly affect the ratio of outward moving actin features (*p*-values; Primary CD4+ vs. Primary control CD8+ cells: 0.566; Primary CD8+ control vs. Jasplakinolide treated cells: 0.392; Primary control CD8+ vs. Latrunculin A treated cells: 1.0; Jasplakinolide vs. Latrunculin treated CD8+ cells: 0.116) (No. of cells analyze $n \geq 5$ cells) (Data represented as Mean ± SEM). (C) The number of detected outward and inward moving features used for analysis in graph B were plotted separately (*p*-values of comparison between inward and outward features: CD4+ cells: 0.017; CD8+ cells: 0.312, Jasplakinolide treated CD8+ cells: 0.013; and Latrunculin treated CD8+ cells: 1.0). (Data represented as Mean ± SEM). (D) The speed of actin features in CD4+ and CD8 + T cells with and without treatment with Jasplakinolide or Latrunculin (*p*-values; Primary CD4+ vs. Primary control CD8+ cells: 0.207; Primary control CD8+ vs. Jasplakinolide treated cells: 0.035; Primary control CD8+ vs. Latrunculin-treated cells: 0.048, Primary CD8+ Jasplakinolide vs. Latrunculin treated cells: 0.033). (Data represented as Mean ± SEM) (E). The speed of outward and inward moving actin features from the data used in Graph D were plotted (*p*-values of comparison between inward and outward speeds are; CD4+ cells: 0.068; control CD8+ cells: 0.062; Jasplakinolide treated CD8+ cells: 0.275; and Latrunculin treated CD8+ cells: 0.156). All statistical analysis in (B–E) was using Mann–Whitney non-parametric two-tailed test. (Data represented as Mean ± SEM). Source data are available online for this figure.

