## [Peer Review File · EMBO Reports]

Actin waves guide an outward movement of microclusters in the lymphocyte immunological synapse

Aheria Dey, Samuel Khiangte, Srishti Mandal, Huw York, Marco Fritzsche, Sumantra Sarkar, and Sudha Kumari

Corresponding author(s): Sudha Kumari (Sudhakm@iisc.ac.in) , Sumantra Sarkar (sumantra@iisc.ac.in)

Review Timeline:

Submission Date:	28th May 25
Editorial Decision:	24th Jun 25
Revision Received:	9th Oct 25
Editorial Decision:	10th Nov 25
Revision Received:	21st Nov 25
Accepted:	4th Dec 25

Editor: Achim Breiling

Transaction Report:

Dear Dr. Kumari,

Thank you for the transfer of your manuscript to EMBO reports. I have now received the reports from the three referees that were asked to evaluate your study, which can be found at the end of this email.

As you will see, the referees think that these findings are of interest. However, they have several comments, concerns, and suggestions, indicating that a major revision of the manuscript is necessary to allow publication of the study in EMBO reports. As the reports are below, and all the referee concerns need to be addressed, I will not detail them here.

Given the constructive referee comments, I would like to invite you to revise your manuscript with the understanding that the concerns of the referees must be addressed in the revised manuscript and/or in a detailed point-by-point response. Acceptance of your manuscript will depend on a positive outcome of a second round of review. It is EMBO reports policy to allow a single round of revision only and acceptance of the manuscript will therefore depend on the completeness of your responses included in the next, final version of the manuscript.

- 1) a .docx formatted version of the final manuscript text (including legends for main figures, EV figures and tables), but without the figures included. Figure legends should be compiled at the end of the manuscript text.
- 2) individual production quality figure files as .eps, .tif, .jpg (one file per figure), of main figures and EV figures. Please upload these as separate, individual files upon re-submission.

- 4) a complete author checklist, which you can download from our author guidelines (<https://www.embopress.org/page/journal/14693178/authorguide>). Please insert page numbers in the checklist to indicate where the requested information can be found in the manuscript. The completed author checklist will also be part of the RPF.

- 5) that primary datasets produced in this study (e.g. RNA-seq, ChIP-seq, structural and array data) are deposited in an

appropriate public database. If no primary datasets have been deposited, please also state this in a dedicated section (e.g. 'No primary datasets have been generated and deposited'), see below.

The accession numbers and database should be listed in a formal "Data Availability" section that follows the model below. This is now mandatory (like the COI statement). Please note that the Data Availability Section is restricted to new primary data that are part of this study. This section is mandatory. As indicated above, if no primary datasets have been deposited, please state this in this section

Data availability

6) We now request the publication of original source data with the aim of making primary data more accessible and transparent to the reader. You will receive a separate email with instructions for providing source data with your revised manuscript, including information how to upload and organize the files.

8) Regarding data quantification and statistics, please make sure that the number "n" for how many independent experiments were performed, their nature (biological versus technical replicates), the bars and error bars (e.g. SEM, SD) and the test used to calculate p-values is indicated in the respective figure legends (also for EV and Appendix figures). Please also check that all the p-values are explained in the legend, and that these fit to those shown in the figure. Please provide statistical testing where applicable. Please avoid the phrase 'independent experiment', but clearly state if these were biological or technical replicates. Please also indicate (e.g. with n.s.) if testing was performed, but the differences are not significant. In case n=2, please show the data as separate datapoints without error bars and statistics. See also: <http://www.embopress.org/page/journal/14693178/authorguide#statisticalanalysis>

9) Please add scale bars of similar style and thickness to microscopic images, using clearly visible black or white bars (depending on the background). Please place these in the lower right corner of the images themselves. Please do not write on or near the bars in the image but define the size in the respective figure legend.

10) Please also note our reference format:

12) We now use CRedit to specify the contributions of each author in the journal submission system. CRedit replaces the author contribution section. Please use the free text box to provide more detailed descriptions and do NOT provide your final manuscript text file with an author contributions section. See also our guide to authors: <https://www.embopress.org/page/journal/14693178/authorguide#authorshipguidelines>

13) All Materials and Methods need to be described in the main text using our 'Structured Methods' format, which is required for

all research articles. According to this format, the Methods section should include a Reagents and Tools Table (listing key reagents, experimental models, software, and relevant equipment and including their sources and relevant identifiers), uploaded as separate file, and a Methods section in which we encourage the authors to describe their methods using a step-by-step protocol format with bullet points, to facilitate the adoption of the methodologies across labs. More information on how to adhere to this format as well as downloadable templates (.doc) for the Reagents and Tools Table can be found in our author guidelines (section 'Structured Methods'):

14) Please order the manuscript sections like this, using these names:

Title page - Abstract - Keywords - Introduction - Results & Discussion - Methods - Data availability section - Acknowledgements - Disclosure and Competing Interests Statement - References - Figure legends - Expanded View Figure legends

15) Please make sure that all the funding information is also entered into the online submission system and that it is complete and similar to the one in the acknowledgement section of the manuscript text file.

Finally, please note that all corresponding authors are required to supply an ORCID ID for their name upon submission of a revised manuscript. Please find instructions on how to link the ORCID ID to the account in our manuscript tracking system in our Author guidelines: <http://www.embopress.org/page/journal/14693178/authorguide#authorshipguidelines>

I look forward to seeing a revised version of your manuscript when it is ready. Please let me know if you have questions or comments regarding the revision.

Yours sincerely,

Referee #1:

This manuscript presents a significant and novel discovery regarding the dynamics of T cell receptor (TCR) microclusters and actin waves at the immunological synapse, challenging previous understandings of the F-actin dynamics in this structure. Contrary to the current view, this study identifies a significant pool of TCR microclusters that move anterogradely toward the cell periphery in primary T cells, rather than entirely retrogradely toward the central supramolecular activation cluster (cSMAC). The authors also proposed that the outward movement of TCR microclusters is driven by the actin waves, and the Wiskott-Aldrich Syndrome Protein (WASP) is facilitating the coupling of TCR microclusters to these anterograde actin waves.

This manuscript effectively integrates TIRF-SIM microscopy and computational modeling techniques. However, additional direct evidence is needed to fully confirm that the actin wave drives TCR microcluster transport.

Major comments:

1. While a visual comparison of manual and automated tracking is provided in Supplementary Figure 1, a quantitative analysis comparing the two methods is missing and should be included.
2. In Fig 1F, the authors showed that the velocity of TCR is similar between Jurka and primary T-cells. However, it was not clear whether the data represent retrograde or anterograde velocity. This should be clearly stated.
3. The main evidence for actin waves driving TCR microclusters anterogradely is their correlated motility and computational modeling. However, these data are mainly correlative; stronger and more direct evidence is needed. For example, pharmacologically inhibiting F-actin polymerization or promoting F-actin depolymerization and examining its effect on TCR microcluster movement in the anterograde direction. Alternatively, one can use a topographically modified surface (e.g., a grooved surface) to compromise actin waves and examine its effect on TCR microcluster movement.
4. CK666 was used to rule out the possibility that Arp2/3 acts as a coupler between TCR microclusters and actin waves. However, preventing the activation of Arp2/3 is different from blocking the interaction between Arp2/3 and TCR microclusters. It is therefore unreasonable to rule out the involvement of Arp2/3 based on the effect of CK666 treatment alone. Along the same line, given the importance of Arp2/3 in F-actin re-arrangement (Gomez et al., 2007), it seems odd that the application of CK666 did not affect the actin flow. The authors should examine retrograde flow in the presence of CK666 to confirm the effectiveness of this inhibitor.
5. The authors used WASP^{-/-} primary T cells and computational modeling to confirm that WASP acts as a coupler between TCR microclusters and actin waves. However, there are numerous WASP-interacting proteins (WIPs), and the possibility of WIPs coupling TCR microclusters and actin waves is very high. This possibility may also explain the somewhat confounding result that WASP is selectively used for coupling TCR microclusters to anterograde but not to the retrograde flow. It is therefore too

immature to claim that the anterograde movement of TCR microclusters was coupled to actin waves through WASP (as shown in the abstract) and should be revised.

Minor comments:

1. What is the ROI in Fig 1B and 1C? This should be stated.
2. Supplemental movie #14 is missing.
3. There are several places where the citation of the figure panels is wrong. For example, on page 6, when the TCR associated with the anterograde flows, it moved outward (green segments in the left graph in Figure 3G); otherwise, it moved inward (gray segments in the left graph in Figure 3F). The authors need to carefully re-examine their manuscript.
4. There are no citations for the supplementary figures within the main body of the text. These should be included.

Referee #2:

This contribution documents anti-retrograde flow of a subset of TCR microclusters in primary CD8+ T cells stimulated by coming into contact with a supported bilayer containing ICAM and anti CD3 antibodies, and attributes this motion to localized anti-retrograde actin wavefronts that couple to TCR microclusters via WASP. Overall, the conclusions appear sound and consistent with the data presented. My main questions/concerns regard the presentation of uncertainty in figures and the main text.

It is stated in the text that microclusters are identified with an accuracy of 97.2% but how this number was determined was not described in the referenced figure and supp fig.

Throughout the text when numbers are presented, they should have associated errors. There are many cases of numbers with 3 or 4 significant figures and no errors in the main text.

When viewing the movies, there is some loss of focus during part of the imaging - how does this contribute to the results presented?

Fig 1 E,G,H, Fig2 B: these display items would be more informative if they included statistical information so that a reader could judge significance (some is included in the caption, but including in the visual would be beneficial). It should be clearly stated if statistics come from cells being independent or clusters being independent. Are results significant with both assumptions? It would be helpful to briefly describe how the "wavefronts" were identified in the main text, since this is an important part of the argument, and these are inferred through a more complicated filtering of the image data. In the methods for this analysis, it would be useful to convert pixel units to actual distances. (e.g. 5x5 gaussian filter, colocalization means that the TCR is within the 8 pixel neighborhood.)

Regarding the model, it is not clear to me what this adds - from a basic understanding it appears that there are enough free parameters to capture whatever desired features in the results, and that the matching of "binding" coefficients is somewhat by design. This seems like more of a cartoon than a predictive model?

Related to the above: If I understand correctly, Jurkat's don't exhibit this anti-retrograde motion because they lack the anti-retrograde actin waves and not necessarily that they couple differently to such waves if they were to be prevalent. How then is it that a binding constant can be estimated? If it is about binding, then a prediction of the model is that WASP over-expression would lead to increased anti-retrograde motion in these cells?

The quality of the actin images for the CK666 and WASP -/- cells in Fig 3 is notably lower than for the images used to draw conclusions for Fig 2 (I also think the movies include some compression artifacts which doesn't help). Is this due to the perturbations or for another reason? How does this impact the analysis?

Velocity is a vector - I think the labels in Fig 1F, 3B should be speed?
The text in Fig 3B is too small to be legible.

The values for primary and Jurkat differ between Figs 2H and 3F, I am guessing because of experimental variation -- Maybe it would be better to plot points for individual cells rather than a single point to better represent confidence?

Referee #3:

This manuscript studies the organisation of the immunological synapse in terms of the movement of antigen-engaged receptor microclusters to form the central supramolecular activation cluster (cSMAC), which is regulated by retrograde F-actin flow. This can be observed in live cells using over-expressed fluorescent constructs such as LifeAct. Actin movement is known to regulate antigen receptor homeostasis at cell-to-cell contact points between T cells and antigen-presenting cells, or in this case lipid

bilayers simulating the surface of these cells. This is achieved by activating the T cell with anti-CD3 epsilon antibodies and by the presence of ICAM-1, which regulates integrin activity. This study uses primary murine CD8 T cells and the human leukaemic Jurkat T cell line to investigate these phenomena. Authors observed an anterograde flow of TCR and actin waves in primary CD8 cells but not in Jurkat T cells. Additionally, anterograde movement of TCR is observed to decrease in WASP-deficient primary CD8 T cells, whereas this event does not seem to be mimicked by Jurkat T cells. This work is too preliminary at this stage, with methodological issues regarding the cell systems used and with no mechanisms that explain the data included in the manuscript.

A major criticism of this work is that it uses primary CD8 T and Jurkat cells to compare different parameters and events.

1. It is unclear whether CD4 and CD8 T primary cells behave similarly with regard to actin waves. The retraction of CD8 T cells, required for serial contact with different target or antigen-presenting cells may be relevant and differential in this context for actin dynamics. Authors should include primary CD4 T cells from same mice in their studies.
2. Anti-murine CD3 epsilon and anti-human CD3 epsilon antibodies do neither demonstrate the same ability to activate the TCR nor the actin dynamics in corresponding cells. In fact, different antibodies recognising and activating human CD3 epsilon do not show the same type of activation. Concentrations of antibodies are also relevant here. Therefore, the systems are not comparable.
3. The authors do not discriminate between human and murine ICAM-1, which do not show cross-reactivity. The illustration at the end of the study (Figure 3 G) does not depict LFA-1 acting in Jurkat cells. Therefore, this can be a methodological problem in the study.
4. These studies do not include co-stimulation, which is known to be relevant for actin dynamics at the immunological synapse.
5. LifeAct alters the actin parameters or produces different results to those obtained by using fluorescent actin protein in this type of study. There is no information about the amount of LifeAct expressed by the cells. Indeed, is this reproduced with other constructs?
6. Studies have been reported on Jurkat mutations that can affect actin dynamics (Gioia et al., BMC Genomics 2018) that could explain different actin movement and dynamics between primary and Jurkat cells.

Minor issues:

1. Authors should pay attention to quote properly in the text the data shown in the Figures (as example, miscitations of Figure 2F and 2G, page 6)
2. Reference section: reference 36 is incomplete ; and reference 59 is missing in the text.

Referee #1:

This manuscript presents a significant and novel discovery regarding the dynamics of T cell receptor (TCR) microclusters and actin waves at the immunological synapse, challenging previous understandings of the F-actin dynamics in this structure. Contrary to the current view, this study identifies a significant pool of TCR microclusters that move anterogradely toward the cell periphery in primary T cells, rather than entirely retrogradely toward the central supramolecular activation cluster (cSMAC). The authors also proposed that the outward movement of TCR microclusters is driven by the actin waves, and the Wiskott-Aldrich Syndrome Protein (WASP) is facilitating the coupling of TCR microclusters to these anterograde actin waves.

This manuscript effectively integrates TIRF-SIM microscopy and computational modelling techniques. However, additional direct evidence is needed to fully confirm that the actin wave drives TCR microcluster transport.

We thank the reviewer for their insightful comments.

Major comments:

1. While a visual comparison of manual and automated tracking is provided in Supplementary Figure 1, a quantitative analysis comparing the two methods is missing and should be included.

As per the suggestion, we compared the manual vs automated velocities and found them to be comparable (manual: 45.72 ± 6.82 nm/sec; automated: 48.26 ± 8.99 nm/sec). We have included this information in the Main text (Page 4).

2. In Fig 1F, the authors showed that the velocity of TCR is similar between Jurkat and primary T-cells. However, it was not clear whether the data represent retrograde or anterograde velocity. This should be clearly stated.

The inward and outward velocities (speed) of the microclusters do not appear to be significantly different; therefore, we had provided average speed of all microclusters. We have now provided an analysis of inward and outward speeds of microclusters as well (Revised Figure G; Main text, page 4).

3. The main evidence for actin waves driving TCR microclusters anterogradely is their correlated motility and computational modeling. However, these data are mainly correlative; stronger and more direct evidence is needed. For example, pharmacologically inhibiting F-actin

polymerization or promoting F-actin depolymerization and examining its effect on TCR microcluster movement in the anterograde direction. Alternatively, one can use a topographically modified surface (e.g., a grooved surface) to compromise actin waves and examine its effect on TCR microcluster movement.

Indeed a topographical “spatial perturbation” is one of the cleanest way to interrogate TCR/actin coupling, as originally demonstrated Jay Groves’ group (DeMond et al., 2008). Investigating directly if such coupling exists in the anterograde flow as well using patterned surfaces could certainly be informative but is technically quite challenging. However, if the average behaviour of TCR ensembles tracked in DeMond et al in murine CD4+ T cells represents a composite of both anterograde as well as retrogradely translocating microclusters, we can assume that even the anterograde actin flow shows frictional coupling to the TCR as well. We examined murine CD4+ T cell actin behaviour at a high temporal resolution and found robust anterograde waves in these cells (new Supplemental figure 3), just like the CD8+ T cells.

Pharmacological perturbation indeed can also generate deeper insights into TCR/anterograde coupling, as the reviewer rightly mentioned. We have now pharmacologically perturbed actin dynamics using Jasplakinolide and Latrunculin A treatment (Varma et al., 2006, Babich et al., 2012), but found that in either treatment, both anterograde as well as retrograde actin flows are reduced (Supplementary figure 4). This requirement of actin dynamics for both flows would make TCR-anterograde coupling interpretation challenging. Thus far, the only perturbation that has shown selective difference in TCR translocation is of WASP deficiency. We will build on it in the future to gain further mechanistic insights.

4. CK666 was used to rule out the possibility that Arp2/3 acts as a coupler between TCR microclusters and actin waves. However, preventing the activation of Arp2/3 is different from blocking the interaction between Arp2/3 and TCR microclusters. It is therefore unreasonable to rule out the involvement of Arp2/3 based on the effect of CK666 treatment alone.

Along the same line, given the importance of Arp2/3 in F-actin re-arrangement (Gomez et al., 2007), it seems odd that the application of CK666 did not affect the actin flow. The authors should examine retrograde flow in the presence of CK666 to confirm the effectiveness of this inhibitor.

We thank the reviewer for pointing us towards the correct interpretation of the data. We have now edited the text to include the correction (Page 9, paragraph 2).

We also agree with reviewer that the effect of CK666 (or the lack of it) on actin flows in our system is surprising. To examine whether the lack of effect on anterograde fraction is because both anterograde and retrograde flows are affected to a comparable extent or, both are not

affected at all, we plotted the flow speed and found that there was a lack of effect of CK666 on either of flows (revised Figure 3C). However, we are positive that the inhibitor was active because we found a reduction in LifeAct intensity at the synapse in treated cells, and synapse area was also found to be reduced compared to DMSO control cells (Figure 3A). Perhaps a compensatory increase in actin polymerization via other nucleation factors such as formins, can explain this effect, esp. since formins play a major role in at least retrograde flow of microclusters (Murugesan et al., 2016). A similar lack of effect of CK666 (and other Arp2/3 complex inhibitors) has been observed in actin retrograde flow in neuronal growth cone, implying upregulation of compensatory actin nucleation pathways (Yang et al., JCB, 2012).

5. The authors used WASP^{-/-} primary T cells and computational modeling to confirm that WASP acts as a coupler between TCR microclusters and actin waves. However, there are numerous WASP-interacting proteins (WIPs), and the possibility of WIPs coupling TCR microclusters and actin waves is very high. This possibility may also explain the somewhat confounding result that WASP is selectively used for coupling TCR microclusters to anterograde but not to the retrograde flow. It is therefore too immature to claim that the anterograde movement of TCR microclusters was coupled to actin waves through WASP (as shown in the abstract) and should be revised.

We thank the reviewer for pointing this out. Yes, we do agree that evidence for a direct coupling between WASP and TCR is lacking currently. We have now edited the text (Page 9) to amend this conceptual error.

Minor comments:

1. What is the ROI in Fig 1B and 1C? This should be stated.

ROI represents cell boundary. We have now added this information to the figure legend.

2. Supplemental movie #14 is missing.

We are sorry for this error. We have included the video in the revised version of the manuscript.

3. There are several places where the citation of the figure panels is wrong. For example, on page 6, when the TCR associated with the anterograde flows, it moved outward (green segments in the left graph in Figure 3G); otherwise, it moved inward (grey segments in the left graph in Figure 3F). The authors need to carefully re-examine their manuscript.

We have now fixed these errors and have carefully reviewed other figures to ensure accuracy.

4. There are no citations for the supplementary figures within the main body of the text. These should be included.

We have carefully referenced all supplemental movies in the main text now.

Referee #2:

This contribution documents anti-retrograde flow of a subset of TCR microclusters in primary CD8+ T cells stimulated by coming into contact with a supported bilayer containing ICAM and anti CD3 antibodies, and attributes this motion to localized anti-retrograde actin wavefronts that couple to TCR microclusters via WASP. Overall, the conclusions appear sound and consistent with the data presented. My main questions/concerns regard the presentation of uncertainty in figures and the main text.

We are grateful to the reviewer for their insightful suggestions that helped us think more deeply about the cellular phenotypes, refine our quantifications, and generate a better understanding of the working model of anterograde actin flows (see the responses below).

1. It is stated in the text that microclusters are identified with an accuracy of 97.2% but how this number was determined was not described in the referenced figure and supp fig.

Thanks for pointing this out. We have now provided this information in the legend of Supplemental figure 1.

2. Throughout the text when numbers are presented, they should have associated errors. There are many cases of numbers with 3 or 4 significant figures and no errors in the main text.

We have now included the corresponding error values in the text.

3. When viewing the movies, there is some loss of focus during part of the imaging - how does this contribute to the results presented?

Loss of focus during high spatiotemporal imaging is indeed a concern. Keeping this in mind, we had employed an image filtering and analysis scheme to avoid artifacts associated with blurring. We used a combination of gaussian blurring, subtraction, and thresholding (see 'Methods') that

identified and tracked objects even when there was transient loss of focus or contrast (due to focus loss as well sometimes due to photobleaching- a phenomenon common in live fast SIM imaging). Below is a representative figure comparing the TCR tracks in an artificially blurred and unblurred original image, which shows that the TCR tracks and directional fraction is comparable in two image sequences (both 0.52).

4. Fig 1 E, G, H, Fig2 B: these display items would be more informative if they included statistical information so that a reader could judge significance (some is included in the caption, but including in the visual would be beneficial). It should be clearly stated if statistics come from cells being independent or clusters being independent. Are results significant with both assumptions? Provide comparative values because it doesn't

We thank the reviewer for pointing out the error. We have now denoted significance on the plots, and provided explanations of the comparison, as well as the compared entities in figure legends. Also, we find that both the average microcluster direction fraction/ cell, or of all the microclusters pooled across several cells shows a comparable trend of mean values, where ~40% of motile TCR shows anterograde migration, although a statistical analysis of pooled microcluster data, to measure variance and significance, is complex. We have now presented data only assuming cells as independent entities in the revised manuscript.

5. It would be helpful to briefly describe how the "wavefronts" were identified in the main text,

since this is an important part of the argument, and these are inferred through a more complicated filtering of the image data. In the methods for this analysis, it would be useful to convert pixel units to actual distances. (e.g. 5x5 gaussian filter, colocalization means that the TCR is within the 8-pixel neighborhood)

We thank the reviewer for this suggestion that allows clearer detailing of the detection methods. We have now included the wavefront detection scheme in the main text, along with the rolling window size for gaussian filtering in μm .

6. Regarding the model, it is not clear to me what this adds - from a basic understanding it appears that there are enough free parameters to capture whatever desired features in the results, and that the matching of "binding" coefficients is somewhat by design. This seems like more of a cartoon than a predictive model?

We completely agree with the reviewer that focusing only on the binding-unbinding probability and of course because of the several free parameters, it is difficult to draw a clear conclusion, questioning the utility of the model.

We have now substituted that model with data from simulations addressing the association of actin waves with anterograde TCR fraction in deeper detail. We examined whether it is the wavefront of the wave that guides anterograde TCR movement, or the wavefront associated actin flows that guide it. We found that wavefront of the wave alone may be sufficient to alter the direction of TCR trajectory from retrograde to anterograde. The actin flows in the waves may also contribute to TCR anterograde movement, although only within a selective threshold PIV range indicating a speed optimum for anterograde movement of TCR. We have now provided this revised model in Figure 2E-G.

Related to the above: If I understand correctly, Jurkat's don't exhibit this anti-retrograde motion because they lack the anti-retrograde actin waves and not necessarily that they couple differently to such waves if they were to be prevalent. How then is it that a binding constant can be estimated? If it is about binding, then a prediction of the model is that WASP over-expression would lead to increased anti-retrograde motion in these cells?

Indeed, the long-range traveling waves (or actin anterograde flow) we observed in primary T cells are lacking in Jurkats, for reasons we don't yet fully understand. This does make our initial reasoning of TCR- wave uncoupling in the model flawed. We have now removed this analysis and conclusion from the manuscript.

The quality of the actin images for the CK666 and WASP -/- cells in Fig 3 is notably lower than for the images used to draw conclusions for Fig 2 (I also think the movies include some

compression artifacts which doesn't help). Is this due to the perturbations or for another reason? How does this impact the analysis?

We find that LifeAct levels at synapse are lower in WASP-/- or CK666-treated cells. Because of this, the contrast of LifeAct signal is low in the images leading to poorer resolution. However, this alteration does not affect the analysis because of the steps utilized in image processing and feature detection, as described in the previous response to the reviewer's question.

Velocity is a vector - I think the labels in Fig 1F, 3B should be speed?

Indeed. Thanks for pointing this out. We have now substituted 'velocity' for 'speed' in the text.

The text in Fig 3B is too small to be legible.

Fixed now.

The values for primary and Jurkats differ between Figs 2H and 3F, I am guessing because of experimental variation -- Maybe it would be better to plot points for individual cells rather than a single point to better represent confidence?

We thank the reviewer for the suggestion. However, since we are not focusing on the old model now, we have removed the plots of model/experimental comparison altogether.

Referee #3:

This manuscript studies the organisation of the immunological synapse in terms of the movement of antigen-engaged receptor microclusters to form the central supramolecular activation cluster (cSMAC), which is regulated by retrograde F-actin flow. This can be observed in live cells using over-expressed fluorescent constructs such as LifeAct. Actin movement is known to regulate antigen receptor homeostasis at cell-to-cell contact points between T cells and antigen-presenting cells, or in this case lipid bilayers simulating the surface of these cells. This is achieved by activating the T cell with anti-CD3 epsilon antibodies and by the presence of ICAM-1, which regulates integrin activity. This study uses primary murine CD8 T cells and the human leukaemic Jurkat T cell line to investigate these phenomena. Authors observed an anterograde flow of TCR and actin waves in primary CD8 cells but not in Jurkat T cells. Additionally, anterograde movement of TCR is observed to decrease in WASP-deficient primary CD8 T cells, whereas this event does not seem to be mimicked by Jurkat T cells. This work is too preliminary at this stage, with methodological issues regarding the cell systems used and with no mechanisms that explain the data included in the manuscript.

A major criticism of this work is that it uses primary CD8 T and Jurkat cells to compare different parameters and events.

We understand that a major concern is comparison of T cells from two different systems. However, our rationale for using Jurkat cells was to use them as gold standard for monitoring actin behaviour at high spatio-temporal resolution, since they have been extensively utilized in most T cell biology studies previously- also perhaps a reason why the actin waves were previously missed. The crucial essence of our report is not only that actin waves exist, but they exist simultaneously with actin retrograde flow- a phenomenon that has not been observed in any other cellular system to our knowledge. This highlights novel and highly specialized mechanisms that primary T cells utilize during activation.

1. It is unclear whether CD4 and CD8 T primary cells behave similarly with regard to actin waves. The retraction of CD8 T cells, required for serial contact with different target or antigen-presenting cells may be relevant and differential in this context for actin dynamics. Authors should include primary CD4 T cells from same mice in their studies.

We thank the reviewer for this suggestion. We find that CD4+ T cells display actin waves as well, show the same anterograde fraction as CD8+ T cells, and the underlying actin flows travel with the average speed comparable to that of CD8+ T cells. We have included this data in the manuscript (Supplementary figure 4). Use of anterograde waves for retraction of CD8+ T cells is an interesting possibility, although mechanistically hard to envisage, because waves would increase the contact area of the cells with targets and we have not observed a membrane contraction process following waves expansion yet. Still, the waves would likely affect the contact duration with the target, something interesting to explore in the future.

2. Anti-murine CD3 epsilon and anti-human CD3 epsilon antibodies do neither demonstrate the same ability to activate the TCR nor the actin dynamics in corresponding cells. In fact, different antibodies recognising and activating human CD3 epsilon do not show the same type of activation. Concentrations of antibodies are also relevant here. Therefore, the systems are not comparable.

Currently we have utilized two widely investigated T cell activation systems to study immunological synapse biology, using the same concentration of ICAM1 and CD3 agonist antibodies on SLBs. How the qualitative differences in TCR triggering (via antigen dose and affinity) may influence flows fraction and actin waves in general, is indeed a very interesting question, something we have not investigated yet.

3. The authors do not discriminate between human and murine ICAM-1, which do not show cross-reactivity. The illustration at the end of the study (Figure 3 G) does not depict LFA-1 acting in Jurkat cells. Therefore, this can be a methodological problem in the study.

We have used recombinant Human ICAM1 which shows activity for both human and mouse T cell LFA1 (Núñez D. et al., Front Immunol. 2017) and has been for both human and mouse T cells in previous studies.

We apologize for the illustrative error in figure 3G (now F) which may have caused this confusion. In both the cellular systems we have used CD3 agonist antibody and human ICAM-1 and we assume that both TCR and LFA-1 interactions are involved in subsynaptic dynamics in both systems. We have amended this error Figure 3 schematic now.

4. These studies do not include co-stimulation, which is known to be relevant for actin dynamics at the immunological synapse.

Very true. Co-stimulation influences T cell actin dynamics indeed (Wuelfing C et al., 1998; Roybal et al., 2015; Ying Xim Tan et al., 2014). For now, we performed characterization of actin waves using a minimal activation condition for simpler interpretation and since CD28-dependent co-stimulation is not always present in the CD8+ synapse. We do aim to explore the fascinating possibility of a role of co-stimulation in actin wave dynamics in the future.

5. LifeAct alters the actin parameters or produces different results to those obtained by using fluorescent actin protein in this type of study. There is no information about the amount of LifeAct expressed by the cells. Indeed, is this reproduced with other constructs?

Indeed, the use of LifeAct has been called to question in some systems such as budding yeast when expressed at high concentration (Courtemanche et al., 2016), where it was known to alter cytokinetic ring dynamics. We controlled for such artifacts using three logics: first, for most of our experiments, we isolated cells from viable and fertile animals with no apparent developmental defect (Riedl J. 2010). Two, we expanded isolated CD8+ T cells after isolation before experiments, where LifeAct expressing cells showed the same proliferation kinetics as the cells lacking lifeAct. T cell proliferation is a gold standard for monitoring T cell health, and any defect in cytokinesis due to aberrant LifeAct expression would have been visible in these experiments. Third, in primary T cells CK666 treatment is known to show a reduction in LifeAct intensity at the immunological synapse which is comparable to the reduction in phalloidin intensity (Kumari et al., eLife, 2015).

Prior to this study we did test other probes such as siR actin, Utrch and F-tractin to mark synaptic F-actin, and out of the four probes we tested (LifeAct, Utrch, siR actin, and F-tractin), we found LifeAct to represent a distribution most similar to phalloidin (in cells lacking any probe)

at early immunological synapse. Additionally, a murine transgenic line stably expressing either Utchr or F-tractin has not been generated yet. Therefore, we chose stable expression of LifeAct in primary cells derived from LifeAct-GFP expressing mice to mark synaptic actin flows.

6. Studies have been reported on Jurkat mutations that can affect actin dynamics (Gioia et al., BMC Genomics 2018) that could explain different actin movement and dynamics between primary and Jurkat cells.

Thanks for pointing us to this paper. We have now cited it in the discussion section.

Minor issues:

1. Authors should pay attention to quote properly in the text the data shown in the Figures (as example, miscitations of Figure 2F and 2G, page 6).
2. Reference section: reference 36 is incomplete, and reference 59 is missing in the text.

We have amended these issues in the revised manuscript and have carefully checked references as well.

Dear Dr. Kumari,

Thank you for the submission of your revised manuscript to our editorial offices. I have now received the reports from the three referees that I asked to re-evaluate the study, you will find below. As you will see, the referees now support publication of your study in EMBO reports. All three have a few comments and suggestions to improve the manuscript, I ask you to address in a final revised manuscript. Please also provide a final p-b-p-response to these referee points and the editorial requests below.

Editorial requests:

- There are author name discrepancies. It is Kiangte in the manuscript text file vs. Khiantge in the submission system, and Srishti in the manuscripts vs. Srihti in the submission system. Please check and make sure that correct and similar names are used.
- Please provide the e-mail addresses of the corresponding authors on the title page.
- Please provide the abstract written in present tense throughout.
- Please order the manuscript sections like this, using only these names:
Title page - Abstract - Keywords - Introduction - Results & Discussion - Methods - Data availability section - Acknowledgements - Disclosure and Competing Interests Statement - References - Figure legends - Expanded View Figure Legends
- Please provide subheadings for the Results & Discussion section to render it more comprehensive and structured.
- In the rebuttal letter, you mention twice a Supplementary Fig. 4 (which referee #1 refers to in his/her report). Please check and provide the figure if this was missing with your final revised manuscript.
- The nomenclature of the supplementary items is not correct. Please use Figure EVx and Movie EVx in all places (source file names, legends, titles in the system callouts). The legends for the EV figures (presently "Supplementary data") needs to be updated to "Expanded View Figure Legends".
- Please remove the movie legends from the manuscript text file. Each legend should be provided in a separate text file and zipped up with its corresponding movie so that we have 20 zip movie folders uploaded.
- Please add callouts for Movie 12 and Movie 13 (Movie EV12 and Movie EV13).
- Please use our reference format:
<http://www.embopress.org/page/journal/14693178/authorguide#referencesformat>
- Please add scale bars of similar style and thickness to all the images, using clearly visible black or white bars (depending on the background). Please place these in the lower right corner of the images themselves. Please do not write on or near the bars in the image but define the size only in the respective figure legend.
- Please provide a fully completed author checklist, which you can download from our author guidelines (<https://www.embopress.org/page/journal/14693178/authorguide>). Please insert page numbers in the checklist to indicate where the requested information can be found in the manuscript and fill in the author and manuscript information. The completed author checklist will also be part of the RPF.
- Please check again that the number "n" for how many independent experiments were performed, their nature (biological versus technical replicates), the bars and error bars (e.g. SEM, SD) and the test used to calculate p-values is indicated in the respective figure legends (main and Appendix figures). Please also check that all the p-values are explained in the legend, and that these fit to those shown in the figure. Please provide statistical testing where applicable. Please avoid the phrase 'independent experiment' but clearly state if these were biological or technical replicates. Please also indicate (e.g. with n.s.) if testing was performed, but the differences are not significant. In case n=2, please show the data as separate datapoints without error bars and statistics. See also:
<http://www.embopress.org/page/journal/14693178/authorguide#statisticalanalysis>

If n<5, please show single datapoints for diagrams. Moreover:

- Please note that the exact p values are not provided in the legend of figure 1E
- Please note that information related to n is missing in the legends of figures 1E-G
- Please note that the error bars are not defined in the legends of figures 3B, C, E; S2 B; S3 B-E
- Please note that the scale bar needs to be defined for figures S1 A, B
- Please note that the white border is not defined in the legend of figure 3C, S1 B. This needs to be rectified.

- Please note that the green arrows are not defined in the legend of figure 2C, 3C. This needs to be rectified.
- Please make sure that all the funding information is also entered into the online submission system and that it is complete and similar to the one in the acknowledgement section of the manuscript text file. Presently, the Infosys Young Investigator fellowship is only mentioned in the manuscript, whereas the Prime Minister's Research Fellowship (Graduate fellowship) is only mentioned in the submission system. Please check.
- Thanks for providing the source data. Please upload this as one folder per main figure, grouping together all the files for this figure in separate excel files for each panel (and ZIPed together), and one folder for the EV Figures, grouping together all the files for each Figure in separate folders (and ZIPed together). Moreover, please provide a fully completed source data checklist indicating each panel source data has been provided.
- Please add a title to Table 1 that also indicates to which Figure it belongs. It seems this is related to Figure EV3, thus its name should be changed to Table EV1. Please remove this from the main manuscript text file and upload the table (as excel file) separately. Please also update the callout.
- All Materials and Methods need to be described in the main text using our 'Structured Methods' format, which is required for all research articles. According to this format, the Methods section should include a Reagents and Tools Table (listing key reagents, experimental models, software, and relevant equipment and including their sources and relevant identifiers), uploaded as separate file. More information on how to adhere to this format as well as downloadable templates (.doc) for the Reagents and Tools Table can be found in our author guidelines (section 'Structured Methods'):

- Please note that all corresponding authors are required to supply an ORCID ID for their name upon submission of a revised manuscript. We will not proceed with publication if this is not done. The ORCID of Sumantra Sarkar is still missing. Please find instructions on how to link the ORCID ID to the account in our manuscript tracking system in our Author guidelines:
<http://www.embopress.org/page/journal/14693178/authorguide#authorshipguidelines>

In addition, I would need from you uploaded separately (please remove this from the manuscript text file):

Best,

Referee #1:

The authors have address all of my concerns.

However, supplementary figure 4 cannot be found in the revised manuscript and should be provided.

Referee #2:

The authors have addressed my major concerns.

Minor point: Including the errors for the numerical values reported in the text indicates that some values should contain fewer significant digits. Currently, a fixed number of digits is included, but there are cases where the error indicates that fewer should be displayed -- E.g. 45.72 +/- 6.82 should be 45.7 +/- 6.8 or even 46 +/- 7.

Referee #3:

Authors have addressed in the revised manuscript most of my suggestions and concerns either by new experimentation or by discussion.

I have only a remaining suggestion:

Authors should pay attention and correct some of the references:

Ref 21: San Jose, Borroto et al., First author is San Jose.

Ref. 32: Kumari, Curado et al., Please cite properly volume, pages, year.

Ref. 36: Complete this reference. It remained incomplete

Ref. 59: Quote properly, Gioia L, Siddique A, Head SR, Salomon DR, Su AI. A genome-wide survey of mutations in the Jurkat cell line.

BMC Genomics. 2018 May 8;19(1):334

Dear Dr. Kumari,

Thank you for the submission of your revised manuscript to our editorial offices. I have now received the reports from the three referees that I asked to re-evaluate the study, you will find below. As you will see, the referees now support publication of your study in EMBO reports. All three have a few comments and suggestions to improve the manuscript, I ask you to address in a final revised manuscript. Please also provide a final p-b-p-response to these referee points and the editorial requests below.

Editorial requests:

- There are author name discrepancies. It is Kiangte in the manuscript text file vs. Khiantge in the submission system, and Srishti in the manuscripts vs. Srihti in the submission system. Please check and make sure that correct and similar names are used.

- Please provide the e-mail addresses of the corresponding authors on the title page.

Done

- Please provide the abstract written in present tense throughout.

Done

- Please order the manuscript sections like this, using only these names:

Title page - Abstract - Keywords - Introduction - Results & Discussion - Methods - Data availability section - Acknowledgements - Disclosure and Competing Interests Statement - References - Figure legends - Expanded View Figure Legends

Done

- Please provide subheadings for the Results & Discussion section to render it more comprehensive and structured.

Done

- In the rebuttal letter, you mention twice a Supplementary Fig. 4 (which referee #1 refers to in his/her report). Please check and provide the figure if this was missing with your final revised manuscript.

Done

- The nomenclature of the supplementary items is not correct. Please use Figure EVx and Movie EVx in all places (source file names, legends, titles in the system callouts). The legends for the EV figures (presently "Supplementary data") needs to be updated to "Expanded View Figure Legends".

Done

- Please remove the movie legends from the manuscript text file. Each legend should be provided in a separate text file and zipped up with its corresponding movie so that we have 20 zip movie folders uploaded.

Done

- Please add callouts for Movie 12 and Movie 13 (Movie EV12 and Movie EV13).

Done

- Please use our reference format:

Done

- Please add scale bars of similar style and thickness to all the images, using clearly visible black or white bars (depending on the background). Please place these in the lower right corner of the images themselves. Please do not write on or near the bars in the image but define the size only in the respective figure legend.

Done

- Please provide a fully completed author checklist, which you can download from our author guidelines (<https://www.embopress.org/page/journal/14693178/authorguide>). Please insert page numbers in the checklist to indicate where the requested information can be found in the manuscript and fill in the author and manuscript information. The completed author checklist will also be part of the RPF.

Done

- Please check again that the number "n" for how many independent experiments were performed, their nature (biological versus technical replicates), the bars and error bars (e.g. SEM, SD) and the test used to calculate p-values is indicated in the respective figure legends (main and Appendix figures). Please also check that all the p-values are explained in the legend, and that these fit those shown in the figure. Please provide statistical testing where applicable. Please avoid the phrase 'independent experiment' but clearly state if these were biological or technical replicates. Please also indicate (e.g. with n.s.) if testing was performed, but the differences are not significant. In case n=2, please show the data as separate datapoints without error bars and statistics. See also:

<http://www.embopress.org/page/journal/14693178/authorguide#statisticalanalysis>

If $n < 5$, please show single datapoints for diagrams. Moreover:

- Please note that the exact p values are not provided in the legend of figure 1E
- Please note that information related to n is missing in the legends of figures 1E-G
- Please note that the error bars are not defined in the legends of figures 3B, C, E; S2 B; S3 B-E
- Please note that the scale bar needs to be defined for figures S1 A, B
- Please note that the white border is not defined in the legend of figure 3C, S1 B. This needs to be rectified.
- Please note that the green arrows are not defined in the legend of figure 2C, 3C. This needs to be rectified.

Done

- Please make sure that all the funding information is also entered into the online submission system and that it is complete and similar to the one in the acknowledgement section of the manuscript text file. Presently, the Infosys Young Investigator fellowship is only mentioned in the manuscript, whereas the Prime Minister's Research Fellowship (Graduate fellowship) is only mentioned in the submission system. Please check.

Done

- Thanks for providing the source data. Please upload this as one folder per main figure, grouping together all the files for this figure in separate excel files for each panel (and ZIPed together), and one folder for the EV Figures, grouping together all the files for each Figure in separate folders (and ZIPed together). Moreover, please provide a fully completed source data checklist indicating each panel source data has been provided.

Done

- Please add a title to Table 1 that also indicates to which Figure it belongs. It seems this is related to Figure EV3, thus its name should be changed to Table EV1. Please remove this from the main manuscript text file and upload the table (as excel file) separately. Please also update the callout.

- All Materials and Methods need to be described in the main text using our 'Structured Methods' format, which is required for all research articles. According to this format, the Methods section should include a Reagents and Tools Table (listing key reagents, experimental models, software, and relevant equipment and including their sources and relevant identifiers), uploaded as separate file. More information on how to adhere to this format as well as downloadable templates (.doc) for the Reagents and Tools Table can be found in our author guidelines (section 'Structured Methods'):

Done

- Please note that all corresponding authors are required to supply an ORCID ID for their name upon submission of a revised manuscript. We will not proceed with publication if this is not done. The ORCID of Sumantra Sarkar is still missing. Please find instructions on how to link the ORCID ID to the account in our manuscript tracking system in our Author guidelines:

<http://www.embopress.org/page/journal/14693178/authorguide#authorshipguidelines>

Done

In addition, I would need from you uploaded separately (please remove this from the manuscript text file):

Done

Best,

Referee #1:

The authors have address all of my concerns.

However, supplementary figure 4 cannot be found in the revised manuscript and should be provided.

We have amended this error.

Referee #2:

The authors have addressed my major concerns.

Minor point: Including the errors for the numerical values reported in the text indicates that some values should contain fewer significant digits. Currently, a fixed number of digits is included, but there are cases where the error indicates that fewer should be displayed -- E.g. 45.72 ± 6.82 should be 45.7 ± 6.8 or even 46 ± 7 .

Done.

Referee #3:

Authors have addressed in the revised manuscript most of my suggestions and concerns either by new experimentation or by discussion.

I have only a remaining suggestion:

Authors should pay attention and correct some of the references:

Ref 21: San Jose, Borroto et al., First author is San Jose.

Ref. 32: Kumari, Curado et al., Please cite properly volume, pages, year.

Ref. 36: Complete this reference. It remained incomplete

Ref. 59: Quote properly, Gioia L, Siddique A, Head SR, Salomon DR, Su AI.A genome-wide survey of mutations in the Jurkat cell line.

BMC Genomics. 2018 May 8;19(1):334

Done.

Sudha Kumari
Indian Institute of Science
Microbiology and Cell Biology
SA09, IInd floor, Bioscience building
IISc campus, CV Raman road
Bengaluru, KA 560012
India

Dear Dr. Kumari,

I am very pleased to accept your manuscript for publication in the next available issue of EMBO reports. Thank you for your contribution to our journal.

You may qualify for financial assistance for your publication charges - either via a Springer Nature fully open access agreement or an EMBO initiative. Check your eligibility: <https://link.springer.com/journal/44319/how-to-publish-with-us>

Yours sincerely,

>>> Please note that it is EMBO Reports policy for the transcript of the editorial process (containing referee reports and your response letter) to be published as an online supplement to each paper. If you do NOT want this, you will need to inform the Editorial Office via email immediately. More information is available here: <https://link.springer.com/partners/embo-press/editorial-policies#Peer%20review>